# Intestine-enriched *apolipoprotein b* orthologs are required for stem cell progeny differentiation and regeneration in planarians

Lily L. Wong [1], Christina G. Bruxvoort [1,4,5,6], Nicholas I. Cejda [1,7], Matthew R. Delaney[1], Jannette Rodriguez Otero[2,8] & David J. Forsthoefel [1,3✉]

Lipid metabolism plays an instructive role in regulating stem cell state and differentiation. However, the roles of lipid mobilization and utilization in stem cell-driven regeneration are unclear. Planarian flatworms readily restore missing tissue due to injury-induced activation of pluripotent somatic stem cells called neoblasts. Here, we identify two intestine-enriched orthologs of *apolipoprotein b, apob-1* and *apob-2*, which mediate transport of neutral lipid stores from the intestine to target tissues including neoblasts, and are required for tissue homeostasis and regeneration. Inhibition of *apob* function by RNAi causes head regression and lysis in uninjured animals, and delays body axis re-establishment and regeneration of multiple organs in amputated fragments. Furthermore, *apob* RNAi causes expansion of the population of differentiating neoblast progeny and dysregulates expression of genes enriched in differentiating and mature cells in eight major cell type lineages. We conclude that intestine-derived lipids serve as a source of metabolites required for neoblast progeny differentiation.

[1] Genes and Human Disease Research Program, Oklahoma Medical Research Foundation, Oklahoma City, OK, USA. [2] Howard Hughes Medical Institute, Department of Cell and Developmental Biology, University of Illinois at Urbana-Champaign, Urbana, IL, USA. [3] Department of Cell Biology, University of Oklahoma Health Sciences Center, Oklahoma City, OK, USA. [4] Present address: Arthritis and Clinical Immunology Research Program, Oklahoma Medical Research Foundation, Oklahoma City, OK, USA. [5] Present address: Department of Pathology, University of Oklahoma Health Sciences Center, Oklahoma City, OK, USA. [6] Present address: Department of Veteran Affairs Medical Center - Research Services, Oklahoma City, OK, USA. [7] Present address: Center for Biomedical Data Science, Oklahoma Medical Research Foundation, Oklahoma City, OK, USA. [8] Present address: Department of Education, Universidad Interamericana de Puerto Rico, San Juan, Puerto Rico, USA. ✉email: david-forsthoefel@omrf.org

Regeneration requires metabolites and energy for cell proliferation, differentiation, migration, and growth. Currently, specific mechanisms by which various metabolites are produced, transported, and utilized to promote regeneration are not well understood. In animals, neutral lipids (NLs) (e.g., triglycerides and cholesteryl esters) are stored in intracellular lipid droplets, which serve as central organelles for energy and lipid homeostasis[1,2]. Lipid stores can be mobilized either by NL packaging and export in lipoprotein particles (LPs), or by lipolysis and secretion as fatty acids (FAs)[3–5]. Mobilization enables transport to other tissues, for example between intestine, liver, and peripheral tissues such as brain and muscle in vertebrates, or between the fat body, intestine, nervous system, and imaginal discs in *Drosophila*[6–8]. Upon lipase-mediated hydrolysis of NLs, the FAs and cholesterol produced serve as building blocks for membrane biosynthesis, substrates for energy via fatty acid beta oxidation, sources of acetyl-CoA and other precursors for chromatin remodeling, and precursors of signaling molecules like eicosanoids and sphingolipids[9–11].

Considerable evidence indicates that disruption of lipid catabolism, biosynthesis, and transport can dysregulate stem cell pluripotency, proliferation, and differentiation[12–14]. For example, inhibiting fatty acid oxidation (FAO, the process of converting fatty acids to acetyl-CoA) causes symmetric production of differentiating progeny and stem cell depletion in the mammalian hematopoietic lineage[15]. By contrast, in the adult mouse hippocampus, inhibiting FAO promotes exit from quiescence and proliferation of neural stem/progenitor cells in the adult mouse hippocampus[16]. In *Drosophila*, compromising lipolysis and/or FAO causes intestinal stem cell necrosis[17], and loss of germline stem cells[18]. On the other hand, FAO-mediated production of acetyl-CoA (a precursor for histone acetylation) is required for differentiation, but not maintenance, of quiescent hematopoietic stem cells in *Drosophila* larvae, illustrating the growing recognition of the intersection of lipid metabolism and epigenetic regulation[19]. Perturbing lipid synthesis and delivery also dysregulates stem cell state and differentiation dynamics. For example, inhibiting fatty acid synthase (required for de novo lipogenesis) reduces proliferation by adult mouse neural stem and progenitor cells[20]. Similarly, mutations in *hydroxysteroid (17-beta) dehydrogenase 7*, a regulator of cholesterol biosynthesis, cause premature differentiation of neural progenitors during development[21]. In culture, depriving human pluripotent stem cells of extrinsic lipids promotes a "naive-to-primed" intermediate state, demonstrating the importance of exogenous lipid availability for regulation of stem cell state[22].

Because stem and progenitor cell activation, proliferation, and differentiation are central to regeneration, these diverse observations underline the potential importance of lipid metabolism during regenerative growth. Currently, however, evidence that lipid transport and utilization influence regeneration is limited. In zebrafish, *leptin b* (a hormonal regulator of systemic lipid metabolism) is one of the most upregulated genes during fin and heart regeneration[23], and *apolipoprotein E* (a regulator of NL transport in LPs) is upregulated during fin regeneration[24]. In the axolotl *Ambystoma mexicanum*, genes involved in steroid, cholesterol, and fatty acid metabolism are among the most upregulated at later stages of limb regeneration, when differentiation of cartilage and muscle progenitors occurs[25]. In a primary cell culture model of nerve injury, rat retinal ganglion cells regrow projections after axotomy more efficiently in the presence of glial-derived LPs, reinforcing the role of cholesterol in axon regeneration[26]. In mice, *low density lipoprotein receptor* deficiency delays liver regeneration and reduces hepatocyte proliferation[27]. Similarly, elevated FA levels (induced by lipoprotein lipase overexpression) causes lipotoxicity and compromises skeletal muscle regeneration[28],

while inhibition of peroxisomal FAO induces differentiation of myogenic satellite cells and muscle hypertrophy during regeneration[29]. During skin wound repair, inhibiting triglyceride lipase-mediated lipolysis by dermal adipocytes compromises recruitment of inflammatory macrophages and adipocyte fate-switching to extracellular-matrix-secreting myofibroblasts[30].

Although these intriguing observations point to the importance of lipid metabolism, there is scant functional understanding of how lipid transport and utilization influence stem cell regulation during regeneration, particularly in emerging animal models with extensive regenerative capacity. Planarians are freshwater flatworms capable of whole-body regeneration, an ability conferred by pluripotent somatic stem cells called neoblasts that divide and differentiate to replace damaged and lost tissues after amputation[31,32]. Diet-derived NLs are stored in the planarian intestine in lipid droplets, suggesting the intestine is a major lipid storage organ, as in *Drosophila* and *C. elegans*[6,33–35]. Planarian intestinal lipids have been proposed to be a source of metabolites during extended fasting, as well as regeneration[34,36,37]. However, mechanisms by which lipid secretion is controlled, and whether delivery to neoblasts or their progeny is functionally required for regeneration, have not been investigated.

In this study, we show that two intestine-enriched *apolipoprotein b* (*apob*) orthologs are required for NL transport from the intestine to neoblasts and their progeny, and that ApoB function is required for stem cell progeny differentiation and regeneration in the planarian *Schmidtea mediterranea*. In mammals and insects, ApoB and ApoB-like proteins mediate trafficking of NLs in LPs from digestive and storage organs to peripheral target tissues[5,38]. Here, we identify adult stem cells and their differentiating progeny as additional target tissues for ApoB-mediated NL transport in a regeneration competent animal, and propose utilizing planarians as models to understand the influence of lipid metabolism on stem cells during regeneration.

## Results

**Planarian *apolipoprotein b* orthologs are expressed by intestinal cells.** Previously, we demonstrated that knockdown of an intestine-enriched transcription factor, *nkx2.2*, inhibited neoblast proliferation and formation of the regeneration blastema, the unpigmented mass of tissue produced after amputation[39]. These observations suggested that the intestine could play a non-autonomous role in regulating stem cell dynamics. In an effort to identify intestine-enriched transcripts encoding neoblast regulators, we generated transcriptomes from control and *nkx2.2(RNAi)* planarians (Supplementary Fig. 1a–c)[39]. An ortholog of human *apolipoprotein b* ("*apob-1*") encoding conserved Vitellogenin, DUF1943, and von Willebrand Factor D domains, was the second-most-downregulated transcript by significance and fold-change, while a paralog, *apob-2*, was also significantly downregulated (Fig. 1a; Supplementary Fig. 1d, e; Supplementary Data 1a–c and Data 2a, b). Consistent with single cell expression profiling[40] (Supplementary Fig. 1f), both transcripts were highly enriched in intestinal phagocytes, absorptive cells responsible for digestion, nutrient storage, and metabolite secretion (Fig. 1b and Supplementary Fig. 1f–h). In addition, both transcripts were weakly expressed in a small number of cells outside the intestine likely to be differentiating neoblast progeny (Fig. 1b and Supplementary Fig. 1f).

**ApoB orthologs are required for viability and regulate neutral lipid transport.** To determine whether ApoB orthologs were required for homeostatic tissue renewal and/or regeneration, we knocked down *apob-1* and *apob-2* using RNA interference[41]. In uninjured planarians, knockdown of either *apob-1* or *apob-2*

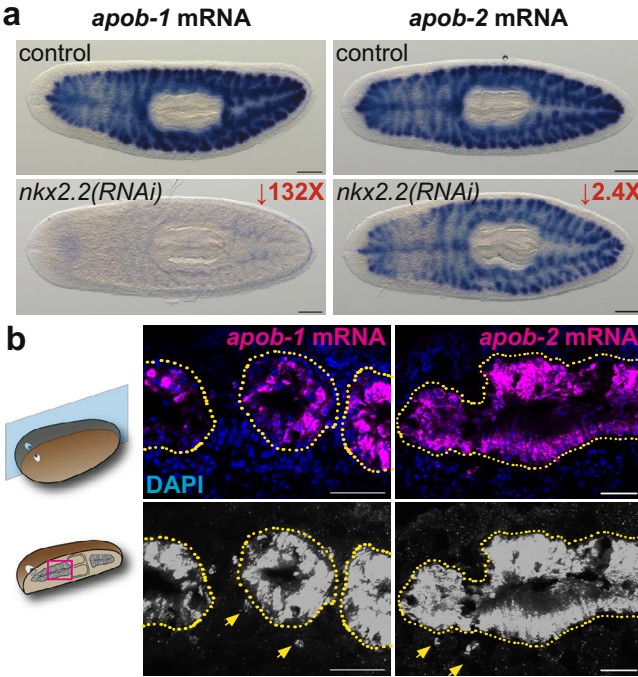

**Fig. 1 Transcripts encoding planarian *apolipoprotein b* orthologs are intestine-enriched and downregulated in *nkx2.2(RNAi)* animals. a** *apob-1* and *apob-2* mRNA (ISH) expression (blue) in control (top) and *nkx2.2(RNAi)* planarians. Images are representative of one experiment with 4/4 animals per condition and probe. **b** *apob-1* and *apob-2* mRNA (FISH) expression (magenta/gray) in sagittal sections. Arrows indicate *apob*-expressing cells (likely differentiating phagocytes) outside the intestine (dotted yellow outline) in digitally brightened images. Images are representative of one experiment with 4/4 animals, and >10 sections per animal per probe. Scale bars: 200 μm (**a**); 50 μm (**b**).

individually had no phenotype (not shown), suggesting functional redundancy. However, after 3–5 double-stranded RNA (dsRNA) feedings, double knockdown *apob(RNAi)* animals displayed phenotypes that progressed from mild (modest/regional pigmentation loss), to severe (animal-wide pigmentation loss and reduced motility), to very severe (head regression, ventral curling, and eventual lysis) (Fig. 2a). For brevity, hereafter we refer to *apob-1;apob-2(RNAi)* double knockdowns as "*apob(RNAi)*", and to specific phenotypic classes as "*apob-M*" for *apob-1;apob-2(RNAi)*-"mild", and "*apob-S*" for *apob-1;apob-2(RNAi)*-"severe". Head regression and ventral curling phenotypes are common in planarians lacking functional neoblasts[42], and suggested that ApoB orthologs could have functional roles in the regulation of planarian stem cells.

ApoB orthologs in vertebrates and insects regulate LP biogenesis and secretion[5,43]. To test whether ApoB protein was secreted, we made a custom antibody recognizing the N-terminus of Smed-ApoB-1 (Fig. 2b, c). ApoB-1 protein was enriched in the intestine, but also throughout the parenchyma (tissues surrounding other organs, sandwiched between the epidermis and intestine, where neoblasts also reside). This indicated that although *apob* mRNAs were intestine-enriched (Fig. 1a, b), ApoB protein was robustly secreted and transported to peripheral tissues (Fig. 2c). Expression was dramatically reduced in both regions in *apob(RNAi)* double knockdown planarians, demonstrating that knockdown effectively reduced ApoB protein levels (Fig. 2c).

ApoB orthologs facilitate NL secretion and transport via LPs[5,38], while ApoB binding by receptors mediate NL uptake and

metabolism by target cells[44]. To test whether planarian ApoB functioned similarly, we first evaluated NL distribution in *apob(RNAi)* animals. In histological sections, NLs (labeled with Oil Red O) were elevated in both the intestine as well as tissues surrounding the intestine, suggesting that both LP secretion by the intestine and uptake/metabolism by peripheral tissues were compromised (Fig. 2d). Using thin layer chromatography, we also found significant elevation of cholesteryl esters and triglycerides in lipid extracts from *apob(RNAi)* animals (Fig. 2e). These phenotypes predicted that delivery of NLs to neoblasts and/or their progeny would be compromised by ApoB inhibition. To test this, we quantified NL content in neoblasts and their progeny using a fluorescent NL probe, BODIPY-493/503, by flow cytometry. Three major subpopulations of planarian cells can be distinguished by their DNA content and sensitivity to X-irradiation[45,46] (Fig. 2f–h). The "X1" fraction/gate includes >2C DNA content neoblasts in S/G2/M phase of the cell cycle, while "X2" includes 2C neoblasts in G0/G1 and G0 post-mitotic progeny. "Xins," named for the fact that cells in this gate are insensitive to X-irradiation, consists of later stage progeny and mature differentiated cells[47]. We found a dramatic reduction of fluorescence in both the X1 and X2 fractions in *apob(RNAi)* animals vs. controls (Fig. 2i, j), indicating that a reduction of NL content in neoblasts and their progeny was caused by ApoB inhibition. Together, these data demonstrated that planarian ApoB proteins were produced by the intestine and were likely secreted as LPs to transport NLs from the intestine to stem cells and their differentiating progeny.

**apob and lipoprotein receptor genes are upregulated in regenerating fragments.** Amputation induces changes in gene expression during planarian regeneration[48–50]. We predicted that if *apob* and other genes involved in neutral lipid metabolism played functional roles during regeneration, they would be up- or downregulated in amputated tissue fragments. Using quantitative PCR, we found that both *apob-1* and *apob-2* transcripts trended upwards in tissue fragments during earlier stages of regeneration commonly associated with neoblast proliferation (1–3 days) and differentiation (2–7 days) (Supplementary Fig. 2a, b)[48,51,52]. Significant upregulation of *apob-1* and *apob-2* was also observed in previously published RNA-Seq data from a 14-day time course of whole-body planarian regeneration (Supplementary Fig. 2c)[50]. Consistently, accumulation of neutral lipids was sustained in 3- and 6-day *apob(RNAi)* regenerates (Supplementary Fig. 2d, e), suggesting that LP trafficking from the intestine was disrupted, as in uninjured animals (Fig. 2d).

In addition, we found that three planarian orthologs of human lipoprotein receptors (which bind apolipoproteins, enabling LP uptake/metabolism) were upregulated in the blastema, and regenerating brain and pharynx (Supplementary Fig. 3a, b, d). These transcripts were expressed in both *piwi-1*-mRNA-positive neoblasts as well as *piwi-1*-negative cells that were likely to be differentiating neoblast progeny in the regenerating pharynx and blastema (Supplementary Fig. 3d). Two lipoprotein receptor genes (*ldlr-1* and *ldlr-2*) were also differentially expressed in the whole-body regeneration RNA-Seq dataset (Supplementary Fig. 3c)[50]. Using the Gene Ontology to mine the same dataset[50], we identified additional planarian genes predicted to regulate lipid transport, triglyceride, and cholesterol metabolism, and found that many of these transcripts were also up- and downregulated during regeneration (Supplementary Fig. 3e and Supplementary Data 3). Together, these observations indicated that *apob-1, apob-2*, and other genes involved in neutral lipid metabolism were differentially expressed in response to amputation, consistent with roles during regeneration.

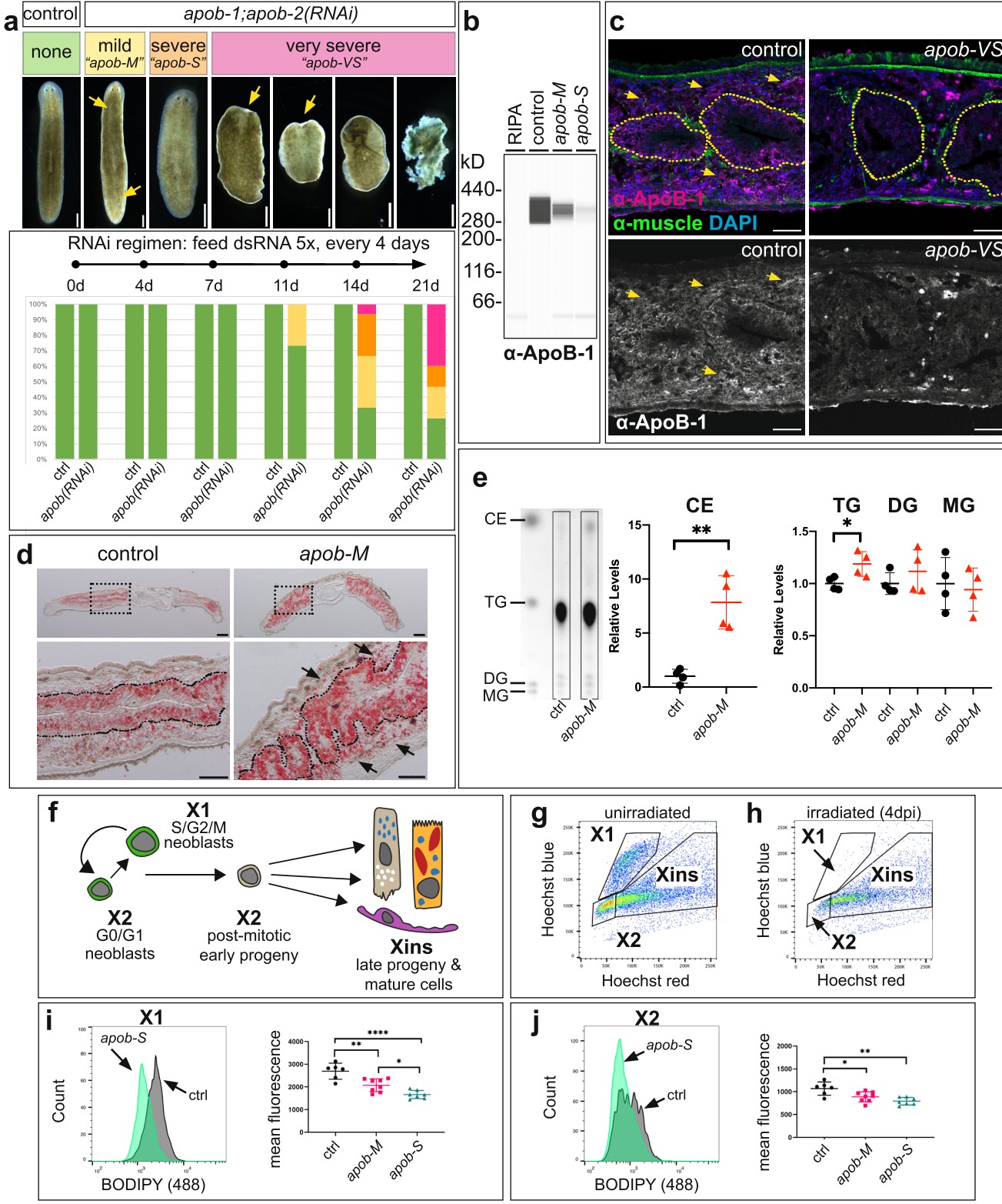

**apob paralogs are required for polarity re-establishment and organogenesis during regeneration.** To assess functional roles of *apob* paralogs during regeneration (Fig. 3a), we chose uninjured animals after three dsRNA feedings that exhibited "mild" and "severe" phenotypes (Fig. 2a), amputated their heads, and assessed blastema morphogenesis, which absolutely requires neoblast proliferation and differentiation[53–55]. Controls, *apob-1(RNAi)*-

only, and *apob-2(RNAi)*-only fragments regenerated normally (Fig. 3b), again suggesting *apob* paralogs likely functioned redundantly. However, *apob(RNAi)* double knockdown animals had reduced posterior blastemas (head fragments) and anterior blastemas (trunk fragments) at 8 days post-amputation (Fig. 3b). Blastema size was noticeably smaller in fragments from "severe" ("*apob-S*") animals compared to "mild" ("*apob-M*") animals

**Fig. 2 *apob* orthologs are required for viability and neutral lipid metabolism. a** Simultaneous RNAi-mediated knockdown of *apob-1* and *apob-2* caused mild ("*apob-M*") (yellow) and severe ("*apob-S*") (orange) depigmentation, and very severe ("*apob-VS*") (pink) phenotypes including head regression, ventral curling, and lysis. $n = 15$ animals per condition, representative of >10 experiments. **b** Simple Western (capillary-based protein analysis) lane view of extracts from control and knockdown planarians labeled with custom anti-ApoB-1. Representative of one experiment with three biological replicates per condition. Uncropped Wes data are in Source Data. **c** ApoB-1 protein expression (magenta), sagittal sections. Expression inside (dotted outline) and outside the intestine (arrows) was reduced in *apob-1(RNAi);apob-2(RNAi)* planarians (right panels) (7 days after the last dsRNA feeding). mAb 6G10 (green) labeled visceral and other muscle fibers. Representative of four independent labeling experiments, $n = 1$ animal per condition (>8 sections per animal). **d** Neutral lipids (red) accumulated in the intestine (dashed outlines) and parenchyma (arrows) of "mild" *apob(RNAi)* planarians (7 days after last dsRNA feeding). Oil Red O labeling, sagittal sections. Representative of two independent experiments, $n = 1$-2 animals per condition (>10 sections per animal). **e** Cholesteryl esters (CE) and triglycerides (TG), but not diacylglycerides (DG) or monoglycerides (MG) were significantly elevated in "mild" *apob(RNAi)* animals. Thin layer chromatography quantification. Student's *t* test (unpaired, two-tailed); \*\*$p = 0.0017$ (CE) and \*$p = 0.0303$ (TG). Error bars: mean ± S.D., $n = 4$ biological replicates, representative of two independent experiments. Uncropped TLC plate is provided in Source Data. **f** Lineage schematic indicating cell types in X1, X2, and Xins subpopulations. **g** Example flow plot for uninjured animals. **h** Example plot for animals 4 days post irradiation ("4 dpi"), showing ablation of cells in X1 and depletion of cells in X2. **i, j** *apob* knockdown caused reduction of neutral lipids in X1 and X2 cells. One-way ANOVA with Tukey's multiple comparisons test. Error bars: mean ± S.D., $n = 6$ (control), 8 (*apob-M*), or 7 (*apob-S*) biological replicates per condition. **i** \*$p = 0.0275$, \*\*$p = 0.0018$, \*\*\*\*$p < 0.0001$. **j** \*$p = 0.0216$, \*\*$p = 0.001$. Data are representative of two independent experiments. Scale bars: 50 μm (**c**); 200 μm (**d** upper panels); 100 μm (**d** lower panels).

(Fig. 3b), suggesting that neoblasts and/or their progeny were more strongly affected.

Expression of so-called "position control genes" to re-establish axial polarity is an essential early event required for whole-body regeneration[56]. We asked whether expression of anterior (*notum*)[57] or posterior (*wnt11-2*)[58] was affected in *apob(RNAi)* regenerates. Indeed, at three days post-amputation (3 dpa), the number of both posterior *wnt11-2*-expressing cells (Fig. 3c) and anterior *notum*-expressing cells (Fig. 3d) was reduced in *apob-M* and *apob-S* fragments. By 6 dpa, although most *apob(RNAi)* fragments had regenerated more of these cells, many fragments (especially *apob-S*) still had fewer than five cells compared to control fragments (Fig. 3c, d). These data suggested that *apob* inhibition delayed, but did not completely abolish, re-establishment of these cells.

*apob* RNAi also affected regeneration of planarian organs, including the brain, pharynx, and intestine. Although both brain and pharynx regenerated, these organs were smaller than in controls (Supplementary Fig. 2f, g). This suggested that regeneration was delayed, but not blocked, similar to the phenotype for cells expressing polarity cues. Quantitatively, at 6 dpa, both brain and pharynx were smaller in size, especially in *apob-S* fragments (Fig. 3e, f). To assess intestine, we analyzed both head and tail regenerates, in which a combination of neoblast-driven new cell production and remodeling of pre-existing, differentiated intestinal branches is required for successful regeneration[51]. Newly regenerated posterior branches were significantly shorter in *apob(RNAi)* head fragments (Fig. 3g). By contrast, in tail fragments, length of the new anterior intestine was not significantly affected (Fig. 3h). Rather, these branches failed to fuse at the anterior midline, leading to a "split" anterior branch phenotype in ~50% of *apob(RNAi)* tail fragments (Fig. 3h). Together, these results suggested that ApoB reduction delayed regeneration of multiple cell types and organs, and, in the case of the intestine, affected differentiation of new cells as well as collective cell migration or other processes required for remodeling.

**apob knockdown causes accumulation of early neoblast progeny.** Because *apob* knockdown disrupted multiple neoblast-driven processes during regeneration, we asked whether defects in neoblast maintenance or proliferation were responsible for the effects of *apob* RNAi on regeneration. First, we assessed neoblast numbers by FISH using *piwi-1*, a pan-neoblast marker, and *tgs-1*, a marker for a more pluripotent subpopulation (which also includes neural specialized neoblasts)[50,53,59]. In both uninjured planarians and 7.5-day regenerates, neoblast numbers and distribution were grossly normal (Supplementary Fig. 4a, b). We also

assessed proliferation using anti-phospho-Histone H3 (Ser10) ("PS10") immunolabeling, which marks cells in late G2 and M phase of the cell cycle[60,61]. In uninjured animals and 2-day head regenerates, the number of mitotic neoblasts increased modestly in *apob-M*, but not *apob-S* samples (Fig. 4a, b). In 2-day trunk regenerates, there was a more significant (~50%) increase in *apob-M* fragments, and a modest increase in *apob-S* fragments (Fig. 4b). Together, these results suggested that ApoB reduction might cause moderate hyperproliferation, or, alternatively, a modest mitotic delay, without dramatically affecting *piwi-1+* or *tgs-1+* neoblast numbers. These mild phenotypes also raised the possibility that ApoB reduction might preferentially dysregulate differentiation, rather than proliferation or maintenance of actively cycling neoblasts.

We tested these possibilities quantitatively using live cell flow cytometry. We dissociated uninjured and regenerating planarians, and evaluated the proportions of cells in the X1 and X2 fractions (Fig. 4c). In uninjured planarians, as well as 2-day and 7-day regeneration fragments, the relative X1 neoblast cell fraction was unaffected by *apob* knockdown (Fig. 4d, e and Supplementary Fig. 4c), consistent with the moderate effect of *apob* RNAi on neoblast proliferation (Fig. 4a, b and Supplementary Fig. 4a, b). By contrast, in uninjured planarians as well as 2- and 7-day trunk fragments, the proportion of cells in the X2 gate increased significantly by ~20–40% (Fig. 4f and Supplementary Fig. 4c) in *apob(RNAi)* animals vs. control. In 2 dpa head fragments, there was also a modest, but statistically insignificant, increase (~15%) in the X2 fraction (Fig. 4f). Because the X2 fraction includes both cycling G1-phase neoblasts and differentiating post-mitotic progeny[47,62], these results suggested that *apob* inhibition might cause lengthening of G1 phase of the cell cycle, and/or a delay in differentiation of neoblast progeny, either of which could increase the proportion of cells in X2.

In order to distinguish between these possibilities, we examined the X2 fraction in uninjured planarians 24 h after X-irradiation, which preferentially ablates over 95% radiation-sensitive cycling neoblasts, without affecting many early neoblast progeny[47,50,55]. As expected, the X1 fraction was almost completely eliminated in both control and *apob(RNAi)* samples (Fig. 4g, h). However, although irradiation reduced the proportion of X2 cells by ~48–56% in both control and *apob(RNAi)* animals (Fig. 4i), the expansion of the X2 fraction persisted in *apob-M* (~62.5% increase) and *apob-S* animals (~42.5% increase) relative to controls (Fig. 4i). This increase in radiation-insensitive cells in X2 strongly suggested that the primary defect in *apob* knockdown animals was a delay in differentiation of neoblast progeny.

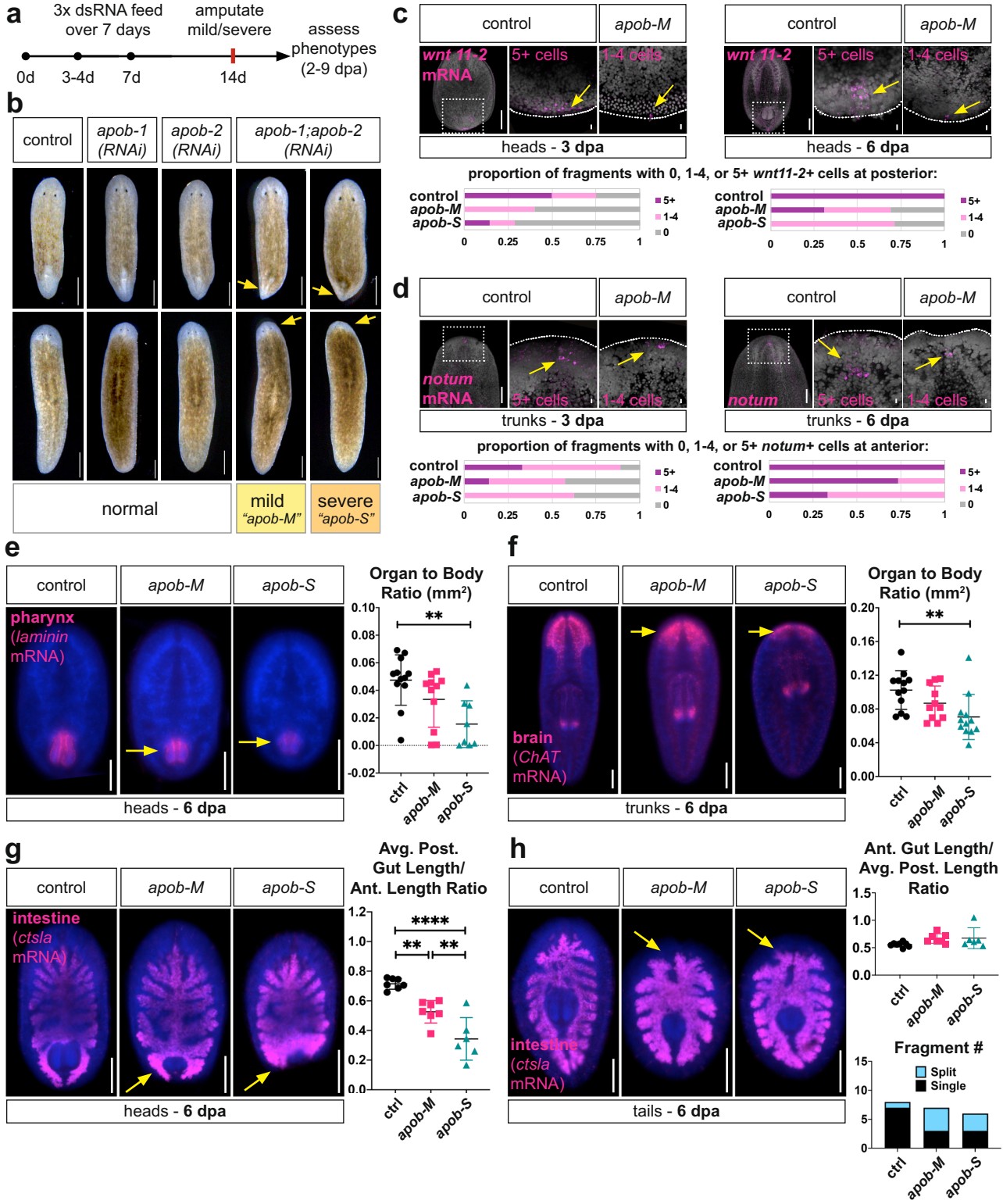

Intriguingly, the increase in radiation-resistant X2 cells was transient: by 48 h post-irradiation, the proportion of X2 cells in *apob(RNAi)* samples was only modestly (but not significantly) elevated (Fig. 4i). This suggested that differentiation was delayed, but not arrested, in *apob(RNAi)* animals. Specifically, at later time points, progeny had differentiated further, and resided in the Xins gate (rather than the X2 gate) as they achieved a more mature cell state. Alternatively, some early progeny might have undergone apoptosis or necrosis at later time points, reducing the apparent accumulation of neoblast progeny in X2.

**ApoB reduction preferentially dysregulates expression of transcripts enriched in differentiating neoblast progeny and mature cell types.** Our flow cytometry data suggested that the primary phenotype in uninjured and regenerating *apob(RNAi)*

**Fig. 3 *apob* inhibition delays regeneration. a** Schematic of RNAi treatment and analysis regimen. **b** *apob* double knockdown resulted in smaller blastemas (arrows) in head and trunk fragments (8 dpa) from "mild" (*apob-M*) and "severe" (*apob-S*) animals. $n = 15$ animals per condition, representative of >10 experiments. **c, d** Differentiation of cells expressing polarity genes *wnt11-2* (posterior) and *notum* (anterior) was delayed in *apob(RNAi)* animals. mRNA FISH (magenta); max projections of confocal images. *wnt11-2* (3 dpa/6 dpa heads): $n = 8/5$ control; 5/13 *apob-M*; 7/7 *apob-S* biological replicates. *notum* (3 dpa/ 6 dpa tails): $n = 9/7$ control; 7/15 *apob-M*; 8/6 *apob-S* biological replicates. Control and *apob-M* fragments shown as phenotype examples. Data representative of two independent experiments. **e** Pharynx regeneration was reduced in *apob(RNAi)* head regenerates (6 dpa) (arrows) (*laminin* mRNA FISH, magenta, epifluorescent images). One-way ANOVA with Tukey's multiple comparisons test. Error bars: mean ± S.D.; $n = 12$ (control), 11 (*apob-M*), or 8 (*apob-S*) biological replicates. **p = 0.0023. Representative of two independent experiments. **f** CNS regeneration was reduced in *apob(RNAi)* tail regenerates (6 dpa) (arrows) (*ChAT* mRNA FISH, magenta, epifluorescent images). One-way ANOVA with Tukey's multiple comparisons test. Error bars: mean ± S.D.; $n = 12$ (control and *apob-S*), or 11 (*apob-M*) biological replicates. **p = 0.0065. Representative of two independent experiments. **g, h** *apob* RNAi disrupted intestine regeneration (6 dpa) (arrows) (*ctsla* mRNA FISH, magenta, epifluorescent images). **g** New branches were shorter in head fragments. One-way ANOVA with Tukey's multiple comparisons test. Error bars: mean ± S.D.; $n = 7$ (control and *apob-M*), or 6 (*apob-S*) biological replicates. **p = 0.0038 (control vs *apob-M*); **p = 0.0065 (*apob-M* vs *apob-S*); ****p = <0.0001. Representative of two independent experiments. **h** New branches often failed to fuse in tail fragments (arrows). $n = 8$ (control), 7 (*apob-M*), or 6 (*apob-S*) biological replicates. Data from one experiment. DAPI in gray (**c, d**), (blue) (**e–h**). Scale bars: 500 µm (**b**); 200 µm (**c, d**); 10 µm (**c, d** insets); 200 µm (**e–h**). Statistical analyses provided in Source Data.

animals was a delay in the differentiation of neoblast progeny. To test this interpretation further, we performed whole animal RNA sequencing on control, *apob-M*, and *apob-S* animals, and identified differentially expressed (DE) genes in the two *apob(RNAi)* groups (Supplementary Fig. 5a–c and Supplementary Data 4). Then, to identify specific biological processes affected by *apob* inhibition, we used the Gene Ontology (GO) to identify patterns of dysregulation in specific functional categories (Supplementary Fig. 5d, e and Supplementary Data 5). In addition, to determine whether *apob* knockdown disproportionately dysregulated genes enriched in differentiating progeny states, we mapped the global *apob* "dysregulation signature" to published bulk- and single-cell transcriptomes to determine which cell types and states were most affected by *apob* inhibition.

*apob* knockdown dysregulated thousands of genes, causing upregulation of 842 (*apob-M*) and 1960 (*apob-S*) transcripts, and downregulation of 1139 (*apob-M*) and 2547 (*apob-S*) transcripts, relative to control (Supplementary Fig. 5a–c and Supplementary Data 4a–d). For upregulated transcripts, "lipid metabolism" was the fifth-most over-represented Biological Process (BP) term in *apob-M* animals, and was the most over-represented category in *apob-S* animals (Supplementary Fig. 5d and Supplementary Data 5a, b). Enrichment of the subcategories of acylglycerol, fatty acid, steroid, and glycerolipid metabolism was consistent with known roles of ApoB orthologs, and suggested a possible compensatory gene expression response to *apob(RNAi)*-induced disruption of NL transport. *apob(RNAi)* also downregulated transcripts in additional metabolism categories, including gluconeogenesis, glycolysis, pyruvate, and nucleotide metabolism (e.g., NADH and ADP), as well as ion transport, indicating wide-ranging dysregulation of metabolite processing and trafficking, especially in *apob-S* animals (Supplementary Fig. 5e and Supplementary Data 5c, d). Interestingly, *apob* inhibition dysregulated many additional, non-metabolism-related transcripts. Upregulated functional categories included differentiation and/or specification of multiple tissues including epidermis, nervous system, and the eye, while downregulated categories included cilium morphogenesis and function (possibly reflecting disruption of ciliated epidermal and/or protonephridial cell numbers or physiology), muscle morphogenesis and function, and extracellular matrix organization (Supplementary Fig. 5d, e and Supplementary Data 5a–d). Importantly, categories related to cell cycle, mitosis, etc. were not enriched among up- or downregulated transcripts. Together, these data suggested that ApoB reduction affected metabolism, cell/tissue differentiation, and functions of mature cell types, but not processes required specifically for cellular proliferation.

Next, to test whether *apob* disproportionately dysregulated genes expressed by differentiating progeny and mature cells, we cross-referenced our DE transcript list with recently published bulk transcriptomes from flow-sorted X1, X2, and Xins cell fractions (Fig. 4c)[50], as well as cells sorted based on their expression of PIWI-1 protein, a widely used marker whose expression marks states within planarian cell lineages (e.g., high PIWI-1 levels in neoblasts, low PIWI-1 levels in some neoblasts and differentiating progeny, and negligible PIWI-1 levels in mature cells)[50,63,64]. We first identified transcripts that were enriched in each cell fraction to generate a "signature" transcript list for each fraction (Supplementary Fig. 6a, c). We then asked what percent of these signature transcripts were dysregulated in *apob-M* and *apob-S* animals (Fig. 5a, b and Supplementary Fig. 6b, d). *apob* RNAi dysregulated signature transcripts in all six fractions. However, compared to X1 or "PIWI-HI" signature transcripts, 2–4 times as many transcripts were up- or downregulated in X2 and "PIWI-LO" cell classes, which included both neoblasts and early progeny (Fig. 5a, b). Relative to Xins and PIWI-NEG fractions, there were also more dysregulated transcripts represented in the X2 and PIWI-LO fractions in *apob* RNAi animals (Fig. 5a, b). To demonstrate the validity of this approach, we cross-referenced transcriptomes from planarians 24 h after irradiation[50], and found that dysregulated transcripts overlapped mainly with X1 and PIWI-HI signature transcripts, and were primarily downregulated, consistent with neoblast loss (Supplementary Fig. 6e, f). Together, this analysis demonstrated that *apob* knockdown preferentially affected transcripts enriched in differentiating progeny, an observation that is consistent with accumulation of cells in this state in flow cytometry experiments (Fig. 4d–i).

To determine whether *apob* RNAi preferentially affected specific lineages or states within individual lineages, we also compared the gene expression signature of *apob(RNAi)* animals with recently published single cell transcriptomes[40]. Specifically, we cross-referenced transcripts dysregulated by *apob* RNAi with transcripts enriched in cell state subclusters in eight planarian lineages (Fig. 5c–i, Supplementary Figs. 8a, c–j and 9a, c, d). The resulting "dysregulation signature" for each lineage showed how *apob* RNAi affected gene expression in specific cell states during the progression from pluripotent cycling neoblasts ("N"), to early transition states ("TS"), to differentiating progeny ("P") to mature cell states ("M") (see example schematic for epidermal lineage in Fig. 5c). In most subclusters, the proportion of downregulated transcripts was greater, but we also observed upregulated transcripts in nearly all subclusters. In addition, the proportion of upregulated transcripts was greater in some protonephridial, *cathepsin*-positive, pharynx, parenchymal, neural, and most intestine subclusters (Fig. 5c–i, Supplementary Figs. 8c–j and 9c, d). This

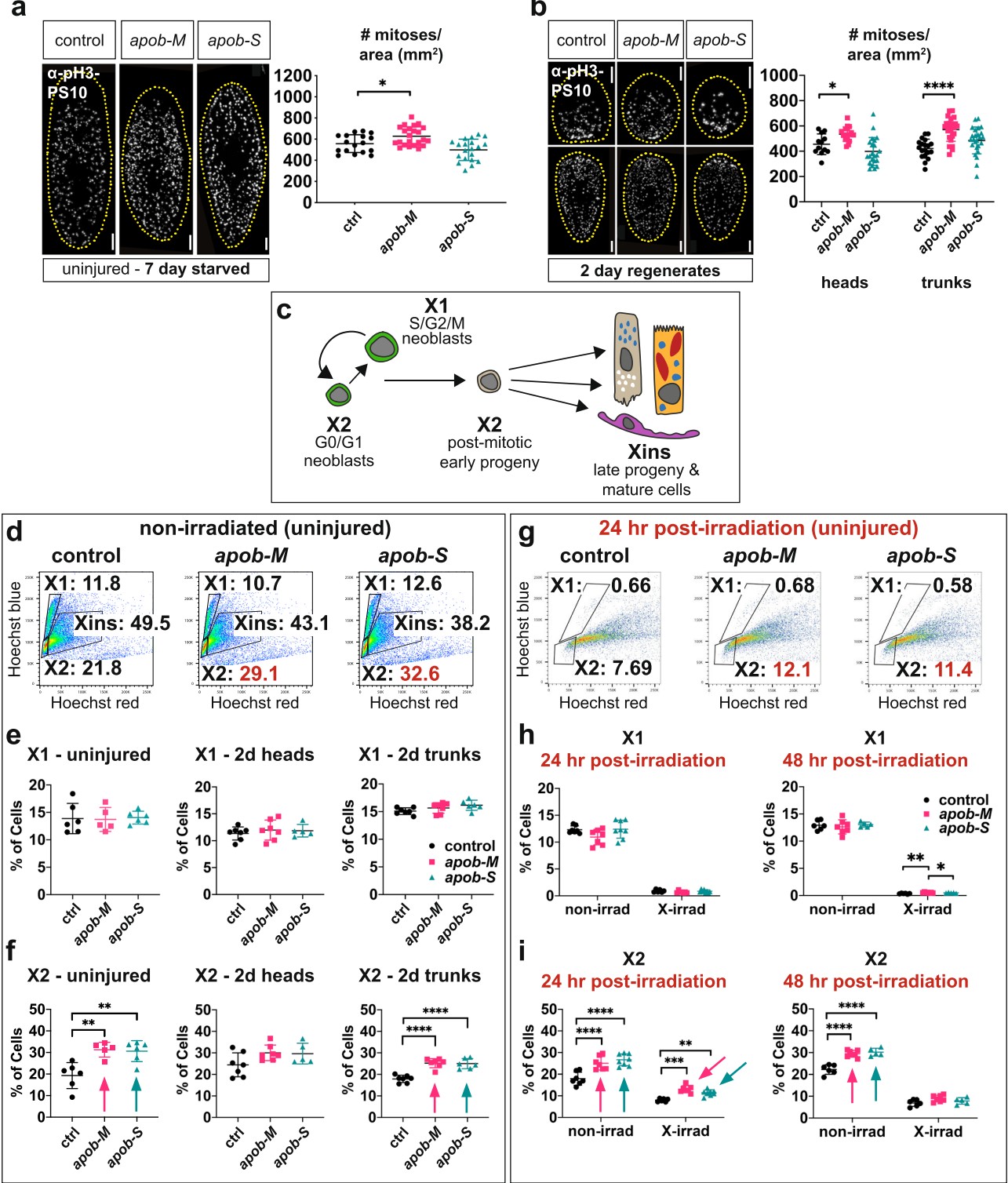

suggested that ApoB inhibition could cause subtle shifts in the proportions of specific cell types, but also that changes in lipid availability could induce gene expression responses (for example, to compensate for lipid accumulation in the intestine).

Despite this complexity, we observed a striking and consistent dysregulation pattern in all eight lineages. In three well-characterized lineages (epidermis, intestine, and protonephridia) (Fig. 5c–i), the percentage of dysregulated "state-enriched" transcripts was lowest for neoblast and early transition state

subclusters, but higher for progeny and mature cell states. Similar trends were observed for less characterized lineages such as muscle, pharynx, and *cathepsin*-positive cells (Supplementary Figs. 8c–j and 9c, d). Validating this approach, irradiation[50] primarily dysregulated neoblast and transition state transcripts (Supplementary Fig. 7a–g), and, to a lesser degree for some lineages, progeny-enriched transcripts (Supplementary Figs. 8a, k–n, and 9a, e). Similarly, and as expected, X1- and PIWI-HI-enriched transcripts overlapped with neoblast and transition state transcripts, while

**Fig. 4 *apob* inhibition causes accumulation of neoblast progeny. a, b** PhosphoHistone-H3-S10-positive (pH3-PS10) cells (white/gray) were modestly elevated in *apob-M* ("mild"), but not *apob-S* ("severe"), uninjured animals (**a**) and 2-day regenerates (**b**). One-way ANOVA with Dunnett's T3 multiple comparisons test. Error bars: mean ± S.D. **a** $n = 17$ (control), 23 (*apob-M*), 20 (*apob-S*) biological replicates. *$p = 0.0304$. **b** Heads: $n = 12$ (control), 17 (*apob-M*), 22 (*apob-S*) biological replicates. *$p = 0.0342$. Tails: $n = 19$ (control), 24 (*apob-M*), 25 (*apob-S*) biological replicates. ****$p < 0.0001$. Representative of two independent experiments. Scale bars: 200 μm. **c** Lineage schematic indicating cell types/states in X1, X2, and Xins subpopulations. **d** Examples of flow dot plots from uninjured planarians indicating percentages of cells in each gate, with X2 increase in *apob(RNAi)* animals in red. Percentage of cells in X1 (**e**) and X2 (**f**) in uninjured (left), 2-day head (middle), and 2-day trunk (right) regenerates. Arrows indicate statistically significant increases in X2. One-way ANOVA with Tukey's multiple comparisons test. Error bars: mean ± S.D, $n \geq 5$ biological replicates per condition. Dots in plots represent exact n values, also provided in Source Data. **$p = 0.004$, ****$p < 0.0001$. Representative of 5 (uninjured) or 2 (regenerates) independent experiments. **g** Examples of flow plots from uninjured planarians, 24 h post-irradiation, indicating percentages of cells in each gate, with X2 increase in *apob(RNAi)* animals in red. Percentage of cells in X1 (**h**) and X2 (**i**) in uninjured planarians, 24 h (left) and 48 h (right) post-irradiation. Arrows indicate significant increases in X2. For X1 and X2 non-irradiated samples, and X1 irradiated samples (**h** and **i**): one-way ANOVA and Tukey's multiple comparisons test (*$p = 0.0211$; **$p = 0.0022$; ****$p < 0.0001$). Differences between RNAi conditions in X1 at 48 h post-irradiation were significant (*$p = 0.0211$; *$p = 0.0022$), but percent of cells in this gate was negligible (<0.7% in all samples). For X2 irradiated samples (**i**): two-way ANOVA and two-stage linear step-up procedure of Benjamini, Krieger and Yekutieli for multiple comparisons (**$q = 0.0049$, ***$q = 0.0004$). Error bars: mean ± S.D, $n \geq 5$ biological replicates per condition. Dots in plots represent exact *n* values, also provided in Source Data. Data from one experiment. Statistical analyses are provided in Source Data.

X2-/PIWI-LO and Xins/PIWI-NEG transcripts were enriched in progressively later cell state subclusters in each lineage (Supplementary Figs. 7h–k, 8b, o–r and 9b, f). Together, these results provided further evidence that *apob* inhibition specifically dysregulated gene expression in progeny and mature cell types/states in all eight lineages, and lent additional support to the interpretation that ApoB was required for differentiation of planarian neoblast progeny.

**apob knockdown delays differentiation of neoblast progeny.** To determine directly whether differentiation of neoblast progeny in *apob(RNAi)* animals was delayed, we generated an anti-PIWI-1 antibody to quantify progeny. As described above, PIWI-1 protein is highly expressed by neoblasts, but perdures at lower levels in early neoblast progeny, before becoming undetectable in late progeny and fully differentiated cells[50,63] (Figs. 5b and 6i). Although PIWI-1 perdurance has not been rigorously analyzed in individual planarian lineages, our bioinformatic analysis suggested that signature PIWI-LO transcripts were highly enriched in early progeny of the well-characterized epidermal lineage (Supplementary Fig. 7i). Accordingly, we analyzed PIWI-1 levels in cells expressing *prog-1* mRNA, a widely used marker of early epidermal progeny, one of the largest progeny subpopulations in planarians[40,47,55,65] (Fig. 6a, b). We predicted that ApoB inhibition might lead to greater numbers of *prog-1*+ cells, and/or a higher proportion of *prog-1*+ cells expressing PIWI-1. The relative proportion of *prog-1*+ cells was similar between *apob(RNAi)* and controls (Fig. 6c), suggesting that lineage specification from pluripotent neoblasts to epidermal lineage-primed progenitor cells to very early progeny was unaffected by ApoB reduction. However, when we counted cells that were both *prog-1*+ and PIWI-LO, we saw a statistically significant increase in this cell fraction in *apob-M* animals, and a modest (but insignificant) increase in *apob-S* animals compared to controls (Fig. 6d). There was also a moderate upward trend in the proportion of *prog-1*+ and PIWI-HI cells (Fig. 6e). In addition, we observed reductions of the *prog-1*+ and PIWI-NEG cell fractions in *apob(RNAi)* animals (significant for *apob-M* animals) (Fig. 6f), indicating a slower rate of PIWI-1 protein degradation, and further supporting the idea that differentiation of epidermal cells was delayed. We reasoned that slowing early progeny differentiation might also cause an eventual reduction in mature cell types. To test this, we quantified the density of mature cells located in the anterior ventral epidermis of *apob(RNAi)* animals (Fig. 6g). Although the number of these cells was unaffected in *apob-M* animals, there

was a significant reduction of epidermal cell density in *apob-S* animals (Fig. 6g). Together, these results directly demonstrated that *apob* knockdown caused a delay in epidermal progeny differentiation, which eventually led to reduced production of mature epidermal cells as the *apob* RNAi phenotype became more severe.

As mentioned above, the correlation of PIWI-1 protein levels with specific states in other planarian lineages has not yet been rigorously characterized. Therefore, instead of analyzing additional lineages, which would require identification of markers that overlap with PIWI-1 in each lineage, we examined cell states by flow cytometry, using low PIWI-1 protein levels as a marker of early progeny, enabling their global quantification (Fig. 6h–k and Supplementary Fig. 10a, b). We detected significant increases in the PIWI-LO cell fraction in both *apob-M* and *apob-S* planarians, compared to controls (Fig. 6h, j). This result suggested that ApoB reduction resulted in longer PIWI-1 protein perdurance in neoblast progeny, and provided further direct evidence of a global, multi-lineage delay in early differentiation. Unexpectedly, we also found statistically significant decreases (~15–20%) in the fraction of PIWI-HI cells in *apob(RNAi)* animals (Fig. 6k). Further analysis of the PIWI-HI subfraction revealed that the percentage of cells with <4C DNA content (i.e., cells in G0/G1 and S) was increased in *apob(RNAi)* animals, while the G2/M fraction was decreased (Supplementary Fig. 10c–f). We observed similar shifting in S-phase and G2/M-phase cell percentages in the X1 (S/G2/M) fraction analyzed earlier (Fig. 4e, Supplementary Fig. 10g–i), despite no detectable overall change in the percentage of cells in this fraction in *apob(RNAi)* animals (Fig. 4e). The increase in PIWI-HI G0/G1 cells (Supplementary Fig. 10d) could reflect earlier dysregulation of PIWI-1 protein reduction (i.e., from high to low levels in very early, recently specified progeny), consistent with the upward trend in *prog-1*+, PIWI-HI cells we observed (Fig. 6e). Additionally, the increase in PIWI-HI G0/G1 and S fractions relative to the G2/M fraction (Supplementary Fig. 10d–f) could indicate a delay in cell cycle progression. This modest reduction in the PIWI-HI cell population could contribute to the delay in *apob(RNAi)* regeneration. However, ApoB inhibition did not dramatically affect the number or location of *piwi-1*+ or *tgs-1*+ neoblasts (Fig. 4a, b) or their gene expression (Fig. 5). Furthermore, cell cycle delays or reduced proliferating neoblasts would be expected to cause a decrease in the proportion of PIWI-LO cells, not the increase we observed. Therefore, we concluded that delay in neoblast progeny

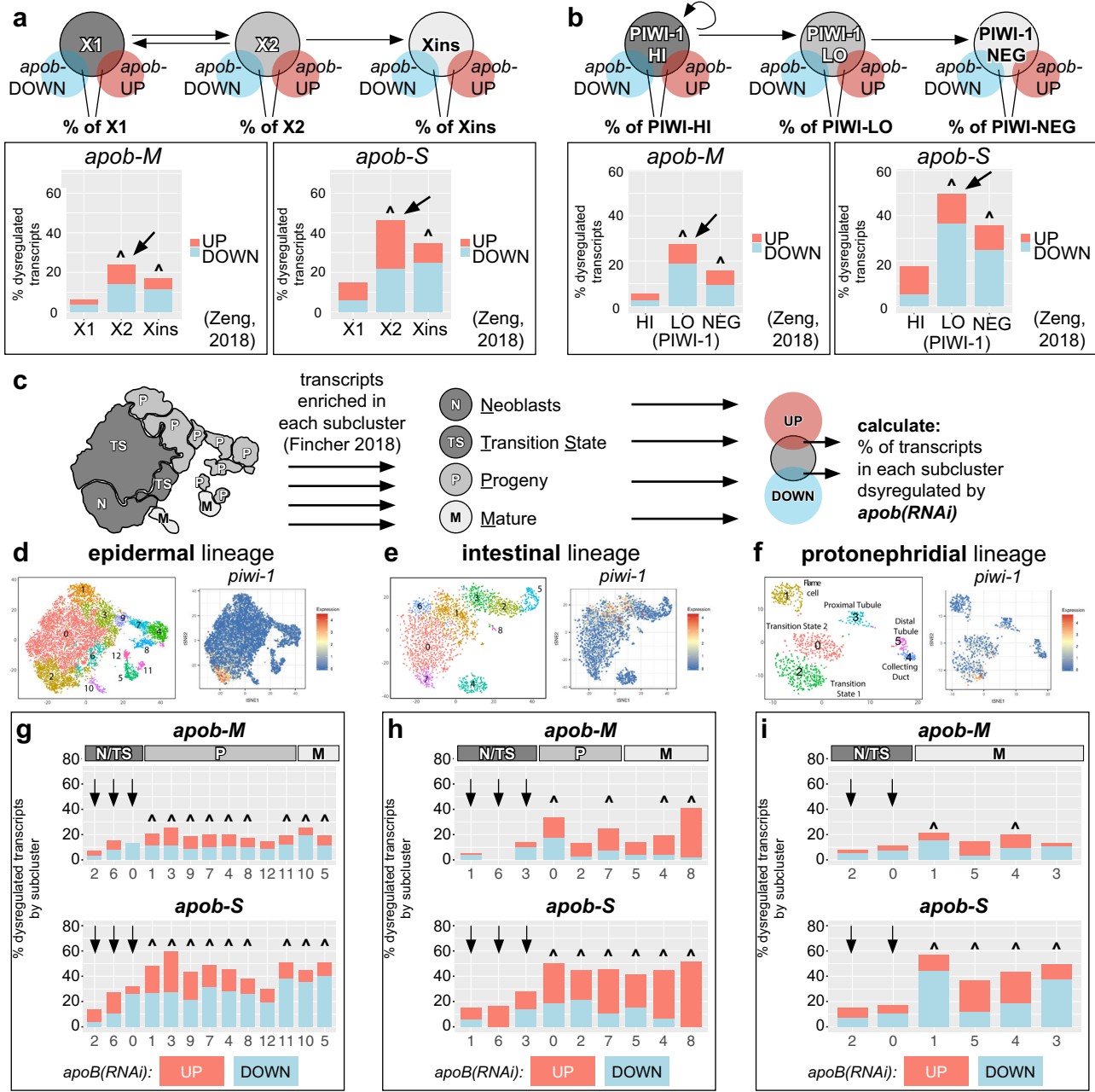

**Fig. 5 *apob* RNAi preferentially dysregulates transcripts in differentiating neoblast progeny and mature cell states.** *apob* RNAi dysregulates greater proportions of X2 (**a**) and PIWI-LO (**b**) signature transcripts (arrows in histograms). Venn diagrams at top show analysis scheme: percent of X1/X2/Xins (**a**) and PIWI-HI/PIWI-LO/PIWI-NEG (**b**) signature transcripts[50] that overlap with transcripts dysregulated in *apob-M* ("mild") and *apob-S* ("severe") animals (this study) (see also Supplementary Fig. 6a–d). Histograms show percentage of signature transcripts up- (red) and downregulated (blue) in *apob-M/apob-S* animals. **c** Schematic example (for epidermal lineage) illustrating how transcripts dysregulated in *apob(RNAi)* animals were cross-referenced with neoblast (N), transition state (TS), progeny (P), and mature (M) cell state subclusters from ref. [40]. See Methods for details. **d–f** t-SNE plots (digiworm.wi.mit.edu) indicate subclusters and *piwi-1* mRNA expression for each lineage. **g–i** *apob* knockdown dysregulated greater proportions of transcripts enriched in progeny ("P") and mature ("M") cell subclusters in multiple cell type lineages. Arrows indicate less-affected transcripts enriched in neoblasts/transition state ("N/TS") subclusters. Carets (ˆ) (**a**, **b**, **g–i**) indicate significant gene expression overlap (*P* < 0.05, Fisher's exact test, see Source Data for individual *p* values).

differentiation was the primary consequence of ApoB inhibition during homeostasis and regeneration.

## Discussion

In this study, we discover a non-cell autonomous role for ApoB proteins in the regulation of planarian neoblast progeny differentiation. Our data support a working model in which ApoB transports NLs from the intestine to both neoblasts and their

differentiating progeny via LPs (Fig. 7). In the absence of ApoB, neoblast proliferation and maintenance are not substantially impaired, but differentiation is slowed, causing an accumulation of neoblast progeny, compromising production of mature cells, and delaying homeostatic cell renewal and regeneration (Fig. 7).

We suggest that ApoB inhibition disrupts the progression of differentiation after cell fate specification, and not specification itself. In planarians, specification likely begins during S-phase,

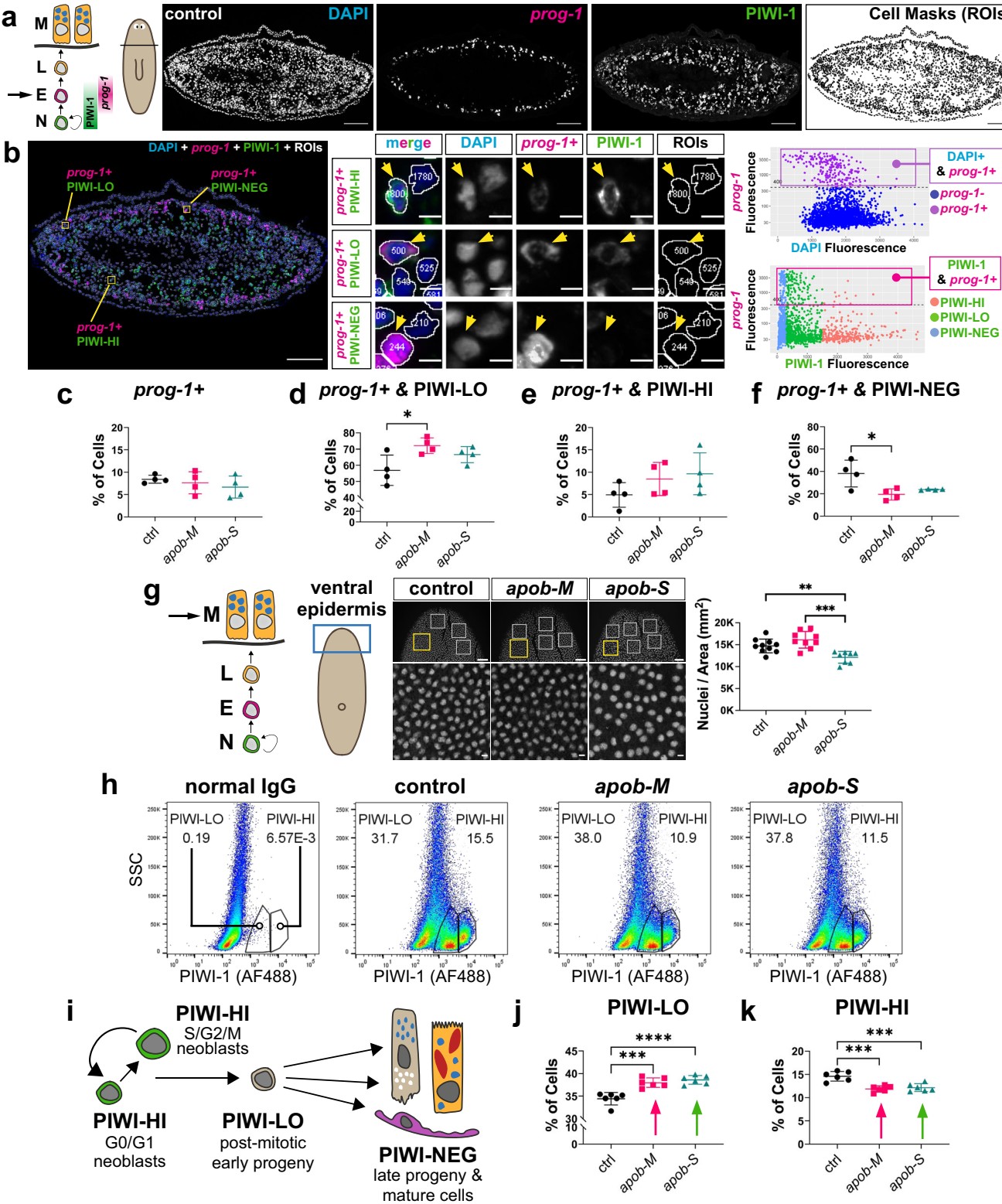

because expression of fate-specific transcription factors is significantly higher in S/G2/M neoblasts than in G1 neoblasts[59,66]. Intriguingly, inhibition of other planarian genes required for differentiation (e.g., the transcription factor *mex3-1*, the extracellular matrix component *collagen4-1*, and the transcriptional co-activating protein *cbp-3*) also cause increases in neoblast numbers in vivo[55,67–69]. Furthermore, knockdown of *exocyst component 3 (exoc3)*, a negative regulator of pluripotency whose

mammalian homolog *Tnfaip3* promotes embryonic stem cell differentiation, causes expansion of the S/G2/M (X1) neoblast fraction[70]. We speculate that these genes may be required for cell fate specification, and that their inhibition shifts neoblast dynamics in favor of renewal divisions that expand the stem cell compartment. By contrast, *apob* RNAi does not cause accumulation of G2/M neoblasts, but rather increases post-mitotic progeny number, and causes greater dysregulation of transcripts

**Fig. 6 Differentiating PIWI-LO neoblast progeny subpopulation increases in *apob(RNAi)* animals. a** Left, schematic of cell state transitions in the epidermal lineage. Cycling neoblasts (N, PIWI-HI); early epidermal progeny (E, PIWI-LO and *prog-1*+); late progeny (L, PIWI-NEG and *prog-1*−); and mature epidermal cells (M, PIWI-NEG and *prog-1*−). Arrow (**a**) indicates early progeny state quantified in (**b**). Right, representative example of anterior transverse section from a control uninjured planarian showing labeling of DAPI (blue), *prog-1* (magenta), PIWI-1 (green), and FIJI-generated cell mask ROIs used for quantification. **b** Left, representative examples of *prog-1*+ cells magnified in middle panels to show PIWI-1 levels (boxes, left and arrows, middle). Upper right, scatter plot indicating DAPI+, *prog-1*+ cells. Lower right, scatter plot indicating DAPI+, *prog-1*+ cells, with PIWI-1 levels colorized. Percentages of *prog-1*+ cells: DAPI+ that are *prog-1*+ (**c**); *prog-1*+ that are PIWI-LO (\**p* = 0.026) (**d**); *prog-1*+ that are PIWI-HI (**e**); and *prog-1*+ that are PIWI-NEG (\**p* = 0.0156) (**f**) in control and *apob(RNAi)* animals. One-way ANOVA and Tukey's multiple comparison test. Error bars: mean ± S.D., *n* = 4 animals (one section per animal, 3000 ± 400 cells per section) per condition. Data from one experiment (**c–f**). **g** Left, epidermal lineage schematic; arrow indicates mature epidermal cells quantified. Middle, examples of DAPI+ nuclei and regions quantified in the anterior ventral epidermis. Right, percentage of nuclei per mm$^2$ area. One-way ANOVA and Tukey's multiple comparison test (\*\**p* = 0.0068, \*\*\**p* = 0.0001). Error bars: mean ± S.D., *n* ≥ 8 biological replicates per condition. Dots in plots represent exact n values, also in Source Data. Data from one experiment. **h** Examples of flow dot plots indicating the gating of PIWI-LO and PIWI-HI cell fractions against side scatter (SSC). **i** Schematic indicating PIWI-1 levels in neoblasts, post-mitotic early progeny, and late progeny/mature cells. Percentages of PIWI-LO (\*\*\**p* = 0.0002; \*\*\*\**p* < 0.0001) (**j**), and PIWI-HI cells (\*\*\**p* = 0.0002 (*apob-M*);\*\*\**p* = 0.0005 (*apob-S*)) (**k**). One-way ANOVA and Tukey's multiple comparison test. Error bars: mean ± S.D., *n* = 6 biological replicates per condition. Representative of two independent experiments. Scale bars: 100 μm (**a**); 100 μm (**b**, left); 5 μm (**b**, right); 50 μm (**g**, upper images), 5 μm (**g**, lower images). Statistical analyses are in Source Data.

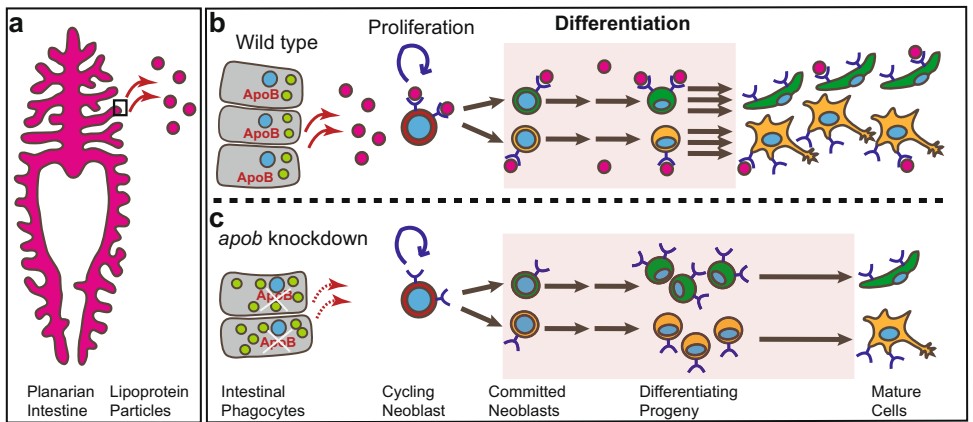

**Fig. 7 A putative model for ApoB function in regulating differentiation of planarian stem cell progeny. a** Our data support a working model in which ApoB is expressed by phagocytes in the intestine, a primary site of LP production and secretion. **b** ApoB mediates secretion of neutral lipids in LPs from intestinal phagocytes to neoblasts and their progeny. **c** In the absence of ApoB, lipids accumulate in the intestine, and LP delivery to neoblasts and their progeny is disrupted, reducing their neutral lipid content. Neoblast proliferation and renewal are largely unaffected by reduced ApoB function. Instead, differentiation and later maturation of most, if not all, planarian cell lineages are slowed, causing an accumulation of differentiating progeny and a delay in regeneration of multiple organs. Box in **a** represents region magnified in **b** wild type and **c** *apob* knockdown conditions. Nuclei, blue. Lipid droplets, light green. LPs, magenta. Apical/lumenal phagocyte surface is to the left and basal/mesenchymal phagocyte surface is to the right in **b** and **c**. Image in **a** reproduced from Forsthoefel et al.[117] under a Creative Commons Attribution 4.0 International License (https://creativecommons.org/licenses/by/4.0/). No changes were made to the original image.

associated with differentiation and mature cell states. Therefore, ApoB is likely required to drive post-specification stages of differentiation, and inhibition delays late commitment steps (i.e., after cell cycle exit), and/or transitions to final mature states (Fig. 7).

Differentiation requires extensive changes in gene expression that are often preceded by genome-wide chromatin remodeling[71,72]. Intriguingly, FA oxidation is a significant source of carbon for acetyl-CoA production and histone acetylation[73]. Consistent with a role for NLs in planarian differentiation, knockdown of *exoc3* reduces triglyceride levels, causes expansion of the neoblast population, inhibits organogenesis, and reduces expression of differentiation markers; palmitic acid supplementation rescues these differentiation-associated phenotypes[70]. Furthermore, in the sexually reproducing *S. mediterranea* planarian biotype, inhibition of *nuclear hormone receptor 1* causes NL accumulation and blocks differentiation of gonads and accessory reproductive tissues, a phenotype that is rescued by supplementation with either acetyl-CoA or Acyl-CoA synthetase[74]. Because acetyl-CoA can also enter the citric acid cycle to produce alpha-ketoglutarate, a substrate for histone

demethylation[75], ApoB inhibition could dysregulate epigenetic changes through multiple pathways. Histone acetylases, deacetylases, methyltransferases, and demethylases are conserved in planarians, and their inhibition disrupts stem cell maintenance, differentiation, and regeneration[68,69,76–79]. Because *apob* RNAi results in widespread dysregulation of thousands of transcripts associated with differentiating progeny, it is reasonable to speculate that in planarians, intestinal lipid stores serve as a ready carbon source that is trafficked by ApoB-containing LPs to neoblasts and progeny to support epigenetic modifications required for differentiation.

ApoB depletion may also delay differentiation by other mechanisms. For example, NL-derived fatty acids may be utilized to produce ATP via beta-oxidation and the mitochondrial electron transport chain (ETC) to support energy-dependent processes during differentiation. Consistent with this idea, planarian mitochondrial mass is higher in differentiating progeny, and pharmacological inhibition of the ETC promotes pluripotency and neoblast colony expansion, which may also limit differentiation[80]. In mammals, disrupting the ETC blocks differentiation of cardiomyocytes and mesenchymal stem cells[81,82],

but the ETC is dispensable for differentiation of mammalian epidermal progenitor cells and *Drosophila* ovarian stem cells[83,84]. Further study will be needed to determine whether LP-transported NLs serve as a significant energy source during planarian differentiation. Additionally, LP-mediated transport of morphogens like Hedgehog or Wnt proteins[85], whose planarian orthologs play important roles in regulating axial polarity and tissue differentiation[56], may be affected by *apob* knockdown. However, we find that ApoB inhibition delays, but does not block or alter regeneration of axial polarity, and we find little evidence of dysregulation of polarity-related transcripts (Supplementary Data 4 and 5), suggesting that planarian LPs may not play a major role in planarian morphogen trafficking. Similarly, fat-soluble vitamins known to influence stem cell dynamics are also transported in LPs[86–91]. Although characterization of LP cargo may yield additional insights, the dramatic dysregulation of lipid metabolism at the gene expression level, and the minimal disruption of vitamin-related gene expression in *apob(RNAi)* animals (Supplementary Data 4 and 5), suggest that LP-mediated vitamin transport may not play a significant role in planarian differentiation. Lastly, we find that *apob* inhibition dysregulated genes associated with muscle differentiation and function (Supplementary Data 5). In planarians, muscle cells not only secrete axial polarity cues, but also serve a fibroblast-like role by secreting most components of the extracellular matrix, whose functions are required to both spatially restrict the stem cell compartment, and modulate proliferation and differentiation[67,92,93]. *apob* RNAi causes moderate downregulation of most fibrillar collagens, as well as the basement membrane *collagen4-1*, which promotes differentiation[67] (Supplementary Data 4 and 5). Thus, ApoB depletion may also delay differentiation indirectly, by compromising the generation and/or function of muscle cells.

The effect of *apoB* knockdown on regeneration suggests several future directions. First, although the existence of prominent LDs in planarian neoblasts has been known for decades[94], the roles of this intriguing organelle have not been investigated. We did not assess neoblast LD numbers or size, but *apob* knockdown dramatically reduces NL content in both X1 (S/G2/M) and X2 (G0/G1) neoblast fractions, suggesting that ApoB and LPs may influence LD content and/or function in both neoblasts and progeny. In addition, NL content is lower in G0/G1 cells, suggesting that a primary role of LDs may be to support differentiation. Once thought of as static storage particles, recent work has demonstrated that LDs in animal cells are dynamic, multifunctional organelles that regulate nutrient sensing, cell stress responses, and even intracellular localization of histones, transcription factors, and other proteins[2,10]. Additionally, although emerging data suggest LDs as a potential therapeutic target in cancer stem cells[95,96], our knowledge of the regulation and functions of LDs in stem cells, especially during regeneration, is limited. Studies in planarians and other regeneration models could further illuminate the roles of this organelle. Second, coordinated metabolic shifts between glycolysis and oxidative phosphorylation may be a widespread aspect of stem cell transitions between quiescence, proliferation, and differentiation, and extrinsic lipids can influence these states[11–14]. Our RNA-Seq data suggest that transcription of regulators of amino acid metabolism, glycolysis, the tricarboxylic acid cycle, and other metabolic pathways respond to *apob(RNAi)*, suggesting that planarian stem cell metabolism is just as dynamic as in other animals. The fact that *apob(RNAi)* seems to primarily affect post-mitotic states also suggests that planarian neoblasts might rely primarily on glycolysis for energy and metabolite supply, and shift to lipid metabolism and oxidative phosphorylation during differentiation, as in other systems[97–99]. Studies in animals with high regenerative capacity could generate greater insights into whether and how

injury can induce metabolic switching. Third, we find that expression of *apob-1* and *apob-2*, as well as numerous other regulators of NL metabolism are dynamically up- and downregulated at the transcript level during regeneration (Supplementary Fig. 3e). This suggests that coordination of lipid metabolism is part of a genome-encoded program of regenerative gene expression. Furthermore, some lipid regulators are up- and downregulated at later regeneration time points (Supplementary Fig. 3e). This raises the intriguing possibility that planarians may replenish NLs utilized during regeneration (Supplementary Fig. 2d, e), or adapt metabolic networks to reduced lipid levels, especially in the absence of post-regeneration feeding. Identification of additional lipid regulators required at different stages of regeneration, and unraveling which transcription factors, chromatin modifiers, and other factors control their expression are thus additional priorities for future study.

Finally, because *apob-1* and *apob-2* are downregulated by inhibition of *nkx2.2*, an intestine-enriched transcription factor also required for regeneration[39], our results provide a specific example of how the planarian intestine can non-autonomously influence neoblast dynamics. Intriguingly, unlike *nkx2.2* RNAi, *apob* knockdown does not reduce the abundance of phosphoHistone-H3-S10-positive neoblasts, indicating that the proliferative defect in *nkx2.2(RNAi)* animals is not caused by *apob* reduction, and additional downstream regulators of proliferation remain to be discovered. Intriguingly, dozens of additional regulators of lipid metabolism and transport of other metabolites are downregulated in *nkx2.2(RNAi)* animals (Supplementary Data 1b), suggesting additional ways in which the intestine could influence neoblasts and their niche.

In summary, we have identified *apolipoprotein B* orthologs and neutral lipid metabolism as important regulators of stem cell progeny differentiation and regenerative tissue growth. Since the discovery of lipoproteins a century ago, their roles in lipid transport and disease have been extensively investigated[100], but functions in stem cell regulation are not nearly as well characterized[101–104]. Efforts to define functions of LPs and their metabolic derivatives more precisely in planarians and other models will therefore improve our understanding of metabolic requirements of stem cell-driven regeneration. In addition, because lipid metabolism is amenable to pharmacological manipulation[105], further study may provide new insights relevant to the dual goals of promoting repair of damaged human tissues, and inhibiting growth in pathological contexts like cancer.

## Methods

**Ethics statement**. Anti-ApoB-1 and anti-PIWI-1 antibodies were generated by GenScript USA (Piscataway, NJ), an OLAW, AAALAC, and PHS-approved vendor. GenScript's animal welfare protocols were approved by OMRF IACUC (17–58). No other vertebrate organisms were used in this study.

**Planarian care**. Asexual *Schmidtea mediterranea* (clonal line CIW4)[106] were maintained in 0.5 g/L Instant Ocean salts (Spectrum Brands SS3-50) with 0.0167 g/L NaHCO₃ (Fisher Scientific S233) dissolved in Type I water[107] for most experiments. For Fig. 6, Supplementary Figs. 10a–f and 11d–f, planarians were maintained in Montjuïc salts (1.6 mmol/l NaCl (Fisher Scientific S271), 1.0 mmol/l CaCl₂ (Sigma 223506), 1.0 mmol/l MgSO₄ (Fisher Scientific M7506), 0.1 mmol/l MgCl₂ (Fisher Scientific BP214), 0.1 mmol/l KCl (Fisher Scientific P217), and 1.2 mmol/l NaHCO₃ prepared in Type II water and UV sterilized)[108,109]. Animals were fed beef or calf liver paste (Sprouts Farmers Market, Oklahoma City). Planarians were starved 7–10 days prior to initiating RNAi. Animals were 2–5 mm in length for most experiments except flow cytometry, for which 5–10 mm animals were used. Uninjured, intact animals were randomly selected from large (300–500 animal) pools.

**Cloning and expressed sequence tags**. Transcripts were cloned as previously described[110]. Sequences were identified in the dd_Smed_v6 transcriptome[111] and the Smed_ESTs3 library[112]. These included *nkx2.2* (dd_2716_0_1/PL08007A2A07), *apob-1* (dd_636_0_1/PL06004B2E09), *apob-2* (dd_194_0_1/PL08004B1B10), *ldlr-1*

(dd_9829_0_1), *ldlr-2* (dd_5596_0_1/PL04021A1C10), *vldlr-1* (dd_1510_0_1/PL05007B1H03), *notum* (dd_24180_0_1), *wnt11-2* (dd_16209_0_1), *choline acetyltransferase/ChAT* (dd_6208_0_1), *laminin* (dd_8356_0_1/PL030015A20A02), *cathepsin La/ctsla* (dd_267_0_1/PL06020B2D09), *piwi-1* (dd_659_0_1/PL06008A2C06), *tgs-1* (dd_10988_0_1), *solute carrier family 22 member 6/slc22a6* (dd_1159_0_1), and Niemann-Pick type C-2/npc2 (dd_73_0_1/PL030001B20C07). *S. mediterranea ldlr-1, ldlr-2,* and *vldlr-1* were identified by BLAST homology and named after their top human refseq_protein BLASTX hits. The *prog-1* (dd_332_0_1) clone was synthesized by Twist Bioscience (South San Francisco, CA). Full sequences of primers, transcripts, and clones, along with accession numbers, are available in Supplementary Data 6.

**Domain organization and phylogenetic analysis**. For ApoB-1, ApoB-2, Ldlr-1, Ldlr-2, and Vldrl-1, protein domains were identified using "HMMSCAN [https://www.ebi.ac.uk/Tools/hmmer/search/hmmscan]" to search Pfam, TIGRFAM, and Superfamily databases, with Phobius for transmembrane and signal peptide predictions (conditional E-value cutoff of 1e-03)[113]. For the ApoB phylogenetic tree, N-terminal Vitellogenin domains from ApoB and related proteins were aligned in Geneious using MAAFT (default settings), and alignment was manually trimmed to the N- and C-terminal boundaries of human Apo B-100. Phylogenetic tree was generated using "PhyML 3.0 [http://www.atgc-montpellier.fr/phyml/]"[114], using AIC for automatic selection of the LG substitution model, BioNJ starting tree, NNI for tree topology improvement, and 100 bootstrap replicates. Accession numbers for proteins used in domain diagrams and phylogenetic analysis are included in Supplementary Data 6.

**In situ hybridization**. Riboprobe synthesis, WISH, and FISH were conducted as previously described[110,115,116]. Briefly, animals were killed by rocking gently (40–50 rpm) in 7.5% N-Acetyl-L-Cysteine (NAc, Sigma A7250-100G) for 15 min at RT, followed by fixation in 4% formaldehyde (EMD Millipore FX0410-5) for 15 min at RT. After dehydration in methanol (−20 °C) (Sigma A412), rehydration, and bleaching in formamide bleaching solution (4 h) (5% formamide (Roche 11814320001), and 1.2% hydrogen peroxide (Sigma H1009), diluted in 0.5X SSC (NaCl - Fisher Scientific S271 and sodium citrate - Fisher Scientific S279)), worms were treated with proteinase K (Invitrogen 100005393, 20 μg/ml) in PBSTx (1X PBS (Invitrogen AM9624), 150 mM NaCl, 0.3% Triton X-100 (Fisher Scientific BP151)) plus 0.1% SDS (Sigma BP8200) (20–30 min for uninjured animals and 5–20 min for regenerates), followed by 10 min post-fixation. For prehybridization, worms were incubated with DIG- (Sigma/Roche 11209256910) or DNP-labeled (Perkin Elmer NEL555001) riboprobes (0.05–1 ng/μl) for 14–16 h in hybridization buffer (56 °C). After post-hybridization washes, worms were incubated with anti-DIG-AP (Roche/Sigma 11093274910, 1:2000), anti-DIG-POD (Roche/Sigma 11207733910, 1:2000) and/or anti-DNP-HRP (Perkin-Elmer FP1129/Akoya Biosciences TS-000400, 1:300) for 14–16 h in TNTx (Tris-HCl pH 8.0 (Invitrogen 15567-027), 150 mM NaCl, 0.3% Triton X-100) with 5% horse serum (Sigma H1138-500ML) and 5% Roche Western Blocking Reagent (Roche/Sigma 11921673001, final 0.5%). After TNTx washes, animals were developed with colorimetric substrate or by tyramide signal amplification (fluor-tyramide at 1:1000–1:2000, 10 min, RT) (tyramides synthesized exactly as described[116]). For double FISH, HRP was inactivated by incubating with 100 mM NaN₃ (Fisher Scientific S227I) for 45 min. DAPI labeling (1–10 μg/ml) (Sigma D9542) was performed either with antibody incubation, or during post-TSA washes, before samples were mounted in Vectashield (Vector Labs H-1000).

Cryosections were generated after FISH (Fig. 1b) or before FISH (Fig. 6a–f) as previously described[117]. On-slide FISH was similar to whole mount FISH, with the following modifications: NAc treatment was 7 min; after sectioning, all steps were performed in polypropylene LockMailers (Ted Pella 21096); and slides were mounted in Fluoromount G (Southern Biotech 0100-01).

**RNAi**. dsRNA synthesis (Thermo Fisher Scientific, MEGAscript T7 Transcription Kit, AM1333) and RNAi experiments were conducted as described[41,110] by mixing in vitro-synthesized dsRNA with 4–9 μl of 1:10 food coloring (Durkee):water mix, and 40 μl of 2:1 liver:water homogenate. For *nkx2.2* RNAi, animals were fed only once with 1 μg *egfp* control or *nkx2.2* dsRNA; RNA was extracted for RNA-Seq after seven days. For *apob* RNAi, 2–6 μg control *egfp* dsRNA or 1–3 μg each *apob-1* and *apob-2* (for simultaneous RNAi) were mixed with liver and food coloring. Animals were fed five times for initial viability experiments, and 3–5 times for most other experiments. Animals with "mild" (*apob-M*) and "severe" (*apob-S*) phenotypes were separated seven days after the last dsRNA feeding prior to fixation, amputation, or flow cytometry of control and *apob(RNAi)* samples. Non-eating planarians were always removed from the experiment if they refused a second dsRNA feeding one day later.

**Anti-ApoB-1 and anti-PIWI-1 antibody generation**. Antigen sequences were supplied to GenScript USA (Piscataway, NJ) for antibody production. For ApoB-1, sequence corresponding to the N-terminal Vitellogenin domain (dd_636, nucleotides 45–1919, amino acids 1–625) was utilized to synthesize expression constructs for fusion protein expression and purification. For PIWI-1, the peptide CVRPKEKTENEPEGP (dd_659, nucleotides 1010–1051, amino acids 302–315, overlapping with a previously used peptide[63]) was used as the antigen. GenScript

immunized New Zealand rabbits, affinity purified the antibody, and assessed antibody titer and immunogenicity with both ELISA and Western blot. The predicted N-terminus of ApoB-1 is only 22% identical and 42% similar to ApoB-2. Furthermore, ApoB-1's mass is predicted to be 235 kD less than ApoB-2. Anti-ApoB-1 labeled only a single prominent band by Western analysis (Fig. 2b and Source Data), suggesting the polyclonal antibody is specific for ApoB-1. Similarly, most cells expressing high levels of PIWI-1 were eliminated 24 h after irradiation (Supplementary Fig. 10a, b), indicating specificity of this antibody for neoblasts and early progeny. Limited quantities of custom antibodies are available upon reasonable request.

**pH3-PS10 immunolabeling**. Mucus removal and fixation were conducted with 2% ice-cold HCl (Sigma H1758) (3 min) and methacarn (60:30:10 methanol:chloroform (Fisher Scientific C298):acetic acid (Fisher Scientific A38-212)) at room temperature (RT, 20 min), followed by bleaching in 6% H₂O₂ in methanol as in ref. [117]. Fixed animals/regenerates were blocked (4 h, RT) in IF block (1X PBS, 0.45% fish gelatin (Sigma G7765-250ML), 0.6% IgG-free BSA (Jackson ImmunoResearch 001-000-162), 0.3% Triton X-100), incubated in rabbit anti-phospho-Histone H3-S10 (Cell Signaling 3377S) at 1:2000 overnight (O/N, 4 °C), washed 8X in PBSTx (1X PBS plus 0.3% Triton X-100) (30 min each, RT), re-blocked for 1 h, then incubated with goat anti-rabbit-HRP (1:2000) (Jackson ImmunoResearch 111-035-003) (O/N, 4 °C). Samples were again washed 8X (20–30 min each, RT), then tyramide signal amplification (TSA) was conducted for 10 min as described[116] with TAMRA-tyramide. Samples were washed in PBSTx for two days, then mounted in Vectashield.

**Immunolabeling (cryosections)**. Cryosections (12 μm) of planarians relaxed in 0.66 M MgCl₂ (Fisher Scientific BP214) and fixed (O/N, 4 °C) in 4% formaldehyde (EM grade, Ted Pella 18505)/1X PBS were generated as described[117]. After rehydration, heat-mediated antigen retrieval (10 min) in 10 mM sodium citrate (Fisher Scientific 6132), pH 6.0 was performed. Sections were permeabilized for 30 min in 1X PBS/0.2% Tween-20 (Fisher Scientific BP337), then blocked for 30 min at RT with 0.45% fish gelatin and 0.6% BSA for 30 min in PBSTw (1X PBS, 0.05% Tween 20). Slides were incubated with custom rabbit anti-ApoB-1 (1:1000, 0.59 μg/ml) and mouse 6G10 anti-muscle (Developmental Studies Hybridoma Bank, 6G10-2C7) (1:250)[118] in blocking buffer (O/N, 4 °C). Slides were washed three times (10 min, RT) with PBSTw after antibody incubation. Slides were then incubated with goat anti-rabbit-HRP (1:2000) (Jackson ImmunoResearch, 111-035-144) and goat anti-mouse-488 (1:250) (Jackson ImmunoResearch, 115-545-146) at RT for 60 min. Slides were washed three times (in PBSTw), with DAPI (2 μg/ml) counterstaining during the first wash. TSA was conducted for 10 min with TAMRA-tyramide as described[116], followed by washes. Slides were mounted in Fluoromount G.

**Cryosection FISH and anti-PIWI-1 labeling**. 16 μm thick cryosections (on SuperFrost Plus slides; Fisher Scientific 12-544-2) from formamide-bleached animals were rehydrated, treated for 5 min at RT with 2 μg/ml proteinase K, and post-fixed. FISH with *prog-1* antisense riboprobe was conducted similarly to whole mount FISH, but in small volume slide mailers (Ted Pella). After post-hybridization TSA, tissue sections were incubated with IF block (60 min, RT), and then with rabbit anti-PIWI-1 (0.5 μg/ml) and mouse 6G10 anti-muscle (1:250) (O/N, 4 °C). Donkey anti-rabbit-Alexa 647 (Jackson 711-605-152) and goat anti-mouse-488 (as above) were used for secondary labeling.

**Epidermal nuclei labeling**. Uninjured planarians were fixed as for FISH (above), then labeled in 10 μg/ml DAPI in PBSTx for four hours at RT. After 4 PBSTx washes, animals were mounted in Fluoromount G.

**Protein extraction and Simple Western analysis**. Five planarians were rocked gently (40–50 rpm) for 7 min in 7.5% N-Acetyl-L-Cysteine and rinsed 2X in 1X PBS. Samples were then homogenized using a motorized Kontes pestle grinder in 250 μl RIPA (50 mM Tris pH 8.0, 150 mM NaCl, 1% NP-40 (Sigma 74385), 0.5% sodium deoxycholate (Sigma D6750), 0.5% SDS) with 40 mM DTT (Sigma D9779) and 1X Halt Protease Inhibitor cocktail (Thermo Scientific 78430). After 30 min on ice, samples were centrifuged (20,817 × g, 15 min, 4 °C), and supernatant was recovered and stored at −80 °C. DTT concentration was reduced by buffer exchange with RIPA (1 mM DTT) using Amicon Ultra 3 kDa columns (UFC500396), then protein concentration was determined using a BCA kit (Pierce 23225) and DeNovix cuvet spectrophotometer according to manufacturers' protocols. For Simple Western (ProteinSimple), lysates were run according to the manufacturer's protocol on a Wes Instrument (running Compass v4.0.0) at 1.0 mg/ml using the 66–440 kDa Wes Separation module (SM-W008) with anti-ApoB-1 at 1:250 (~2.3 μg/ml).

**Oil Red O staining**. Planarians were relaxed in 0.66 M MgCl₂, fixed overnight (RT) in 4% formaldehyde (EM grade) in 1X PBS, protected in sucrose, and cryosectioned (20 μm) onto SuperFrost Plus slides[117]. Slides were rehydrated in deionized (DI) water (3 × 10 min, RT), then stained in Oil Red O (Sigma O0625) solution (6 ml Whatman-filtered 0.5% Oil Red O in 100% isopropanol (Fisher Scientific A416)

plus 4 ml ultrapure water) for 15 min at RT. Slides were quickly dipped 3–5X in 60% isopropanol to remove excess dye, then rinsed for 1 min in 60% isopropanol, then rinsed 1 min in DI water. Slides were then rinsed in PBS-Tween-20 (0.01%, to prevent drying), then mounted in 90% glycerol/1X PBS, and imaged within 2–3 days.

**Thin layer chromatography.** Ten planarians (5–8 mm) were placed in 1.7 ml microcentrifuge tubes with all planarian salts removed, and the animals' mass was obtained. Lipids were then extracted using the Folch method[119]. Briefly, 1 ml ice-cold 2:1 chloroform:methanol was added, then animals were sonicated in ice water in a cup-horn sonicator (10 cycles of 5 s pulses at ~48–55 W). Samples were rocked at RT for 5 h, then centrifuged (2 min at 16,000 × g, 4 °C) to pellet insoluble material. Supernatant was recovered to a new tube and stored at −80 °C. 1 ml 2:1 chloroform:methanol was added to the pellet, re-sonicated, rocked overnight at RT, then centrifuged as above. 0.2 volumes 0.9% NaCl (in water) was added to each extract, tubes were inverted gently 10–15X to mix, vortexed for 10–15 s, then centrifuged (2 min at 2000 × g, RT). Lower phases from each biological replicate (four control and four apob(RNAi) replicates, 10 animals each) were recovered, and speed-vacuumed (30 °C × 60–90 min with spinning) to evaporate solvent. Concentrated lipids were resuspended in 2 μl chloroform per mg animal mass (above) and stored at −80 °C (for less than 7 days). TLC was performed as previously described[120] with slight modifications. 150 Å silica gel HL 250 μl 20 × 20 cm plates (iChromatography 66011/Miles Scientific P76011) were pre-equilibrated with 1:1 chloroform:methanol (~1 h). After drying, 1 μl lipids and 3 μl standards (30 μg mono-, di-, triglyceride mix, SUPELCO 1787-1AMP, plus 30 μg cholesteryl palmitate, SIGMA C6072) were spotted onto the plates using TLC spotting capillaries. Non-polar lipids were resolved with a 80:10:1 petroleum ether (Sigma 77399):ethyl ether (Sigma EX0190):acetic acid mix. After drying, TLC plates were sprayed with Primuline (SIGMA 206865) (1 mg/ml fresh stock in ultrapure water, diluted 1:100 into 80 ml acetone plus 20 ml ultrapure water), dried, and imaged on an Alpha Innotech chemiluminescent imager with Cy2 excitation/emission filters. Peak areas in images were quantified in ImageJ/Fiji using the "Plot Lanes" function in the Gels submenu (https://imagej.nih.gov/ij/docs/menus/analyze.html). Averages were calculated and normalized to controls in Excel.

**RNA extraction, library preparation, and RNA sequencing.** Uninjured planarians (5–6 animals per biological replicate) were homogenized in Trizol (Thermo Fisher Scientific 15596026) using a motorized Kontes pestle grinder, and RNA was extracted using two chloroform extractions and high-salt precipitation buffer according to the manufacturer's instructions. After precipitation, solutions were transferred to Zymo RNA columns (Zymo RNA Clean & Concentrator 5 kit, R1013) for DNAse treatment and purification, according to manufacturer's instructions. RNA samples were analyzed using Agilent RNA ScreenTape (5067–5576) on an Agilent TapeStation 2200 according to manufacturer's protocol.

For analysis of gene expression in control vs. nkx2.2(RNAi) (three biological replicates each), mRNA was enriched using oligo-dT homopolymer beads, and libraries were generated using the Illumina Truseq Stranded mRNA Library Prep Kit according to the manufacturer's protocol. Final libraries were assayed on the Agilent TapeStation for appropriate size and quantity. Libraries were pooled in equimolar amounts as ascertained by fluorometric analysis, then final pools were absolutely quantified using qPCR on a Roche LightCycler 480 with Kapa Biosystems Illumina Library Quantification Reagents. Paired-end (2 × 150 bp) sequence was generated on an Illumina NovaSeq 6000 instrument. 28 M–43 M reads were generated for each of three biological replicates per condition. For analysis of gene expression in control, apob-M, and apob-S animals, total RNA for six (control) or four (apob-M and apob-S) biological replicates per condition was submitted to GENEWIZ (South Plainfield, NJ) for library generation using NEB NEXT ULTRA library prep (New England Biolabs) and RNA sequencing with standard Illumina adapters. Paired-end (2 × 150 bp) sequence was generated on an Illumina HiSeq 4000 instrument; 19 M–26 M reads were generated for each replicate.

**Read mapping.** For both nkx2.2(RNAi) and apob(RNAi) experiments, quality control and read mapping to unique transcripts in dd_Smed_v6[110,111] were conducted with FastQC (v0.11.5)[121], BBDuk (v35.66) (https://sourceforge.net/projects/bbmap/), and Bowtie2 (v2.3.1)[122]. BBDuk (v36.99) settings for paired end reads: k=13 ktrim=r mink=11 qtrim=rl trimq=10 minlength=35 tbo tpe. Bowtie2 (v2.3.1) for paired end reads was used for mapping, with "-a" multi-mapping and "–local" soft-clipping allowed. For read summarization, the "featureCounts" utility in the Subread package (v1.6.3)[123] was used with a custom ".SAF" file and options "-p -M -O -F SAF" to include multi-mapping and multi-overlapping reads.

For mapping of X1/X2/Xins and PIWI-HI/-LO/-NEG bulk sequence, regeneration fragment sequence, and whole animal 24 h post-irradiation sequence[50], fastq files were downloaded from NCBI "GEO GSE107874", mapped to dd_Smed_v6_unique using BBDuk and Bowtie2, followed by count summarization using Samtools as previously described[110].

**Differential expression analysis.** Read counts matrices were imported into R, then analyzed in edgeR v3.26.3 (apob(RNAi) experiment) or v3.16.5 (all others)[124].

First, all transcripts with counts per million (CPM) < 1 in three samples (nxk2.2(RNAi), Zeng X1/X2/Xins data, and Zeng 24 h irradiation data) or four samples (all others) (e.g., lowly expressed transcripts) were excluded from further analysis. Next, after recalculation of library sizes, samples were normalized using trimmed mean of M-values (TMM) method, followed by calculation of common, trended, and tagwise dispersions. Finally, differentially expressed transcripts were identified using the pairwise exact test (nkx2.2(RNAi) and apob(RNAi) experiments, Zeng 24 h irradiation data) or the generalized linear model (GLM) likelihood ratio test (other Zeng datasets). Expression changes were considered to be significant if the false discovery rate-adjusted p value ("FDR") was <0.05. dd_Smed_v6 expression analyses for Zeng data sets are provided in Supplementary Data 7.

**Gene ontology analysis.** GO analysis was conducted using BiNGO[125] (v3.0.4, in Cytoscape 3.8.0) using a custom S. mediterranea GO annotation as previously described[110]. RefSeq protein collections used for BLASTX and UniProtKB Biological Process GO terms used for annotation were downloaded for each organism in April 2020.

**Hierarchical clustering and heat maps.** Hierarchical clustering of transcripts annotated with lipid-metabolism-related GO terms was conducted using EdgeR-generated log2FC values in Cluster 3.0[126], with Euclidean distance and complete linkage. Heat maps were generated with Java Treeview[127].

**qRT-PCR.** Total RNA was extracted from biological triplicates (5–10 fragments per replicate) using Trizol as for RNA-Seq samples. 1 μg total RNA was reverse transcribed using the iScript cDNA Synthesis kit (BioRad 1708890). apob-1 and apob-2 levels were detected using the Fast Start Essential Green DNA master mix (Roche 06924204001) on a Roche LightCycler 96 instrument (SW 1.1 and Light-Cycler 96 Application Version 1.1.0.1320). RNA levels were normalized to the geometric mean of endogenous controls ef-2 and gapdh using the Livak ΔΔCt method[128]. Primer sequences are provided in Supplementary Data 6.

**Flow cytometry.** For live cell flow cytometry, three planarians or regeneration fragments per biological replicate were dissociated and filtered in CMFB with collagenase (Millipore Sigma C5138) as described[66]. Cells were labeled at RT with Hoechst 33342 (Invitrogen H1399) (50 μg/ml) for 45 min, followed by addition of propidium iodide (Sigma P4170) (1 μg/ml). For neutral lipid labeling, BODIPY 493/503 (Molecular Probes D3922) at 10 ng/ml was included with Hoechst. Cells were analyzed on a Becton Dickinson FACSCelesta (BD FACSDiva version 9.0) instrument with 405 nm, 488 nm, and 640 nm lasers. After gating for live cell singlets (Supplementary Fig. 11a–c), X1, X2, and Xins gates were drawn using two criteria: cell proportions were approximately 15% (X1) 25% (X2) and 60% (Xins), and reductions in X1 and X2 fractions in 4-day post-irradiation animals were >95% and ~70%, respectively.

For fixed cell flow cytometry, 4–6 planarians per biological replicate were dissociated as above. Cells were fixed in 2% formaldehyde (EM grade) for 10 min at room temperature (RT), then rinsed twice with 1X PBS. Cells were incubated with permeabilization and blocking buffer (perm/block: 1X PBS, 10% horse serum, 2 mM EDTA (Sigma E5134), 2 mM NaN₃, 0.3% Triton X100) for 30 min. Each sample was split in half to place into 2 wells of a 96-well V bottom plate (~1–2 million cells per well) (ThermoFisher Scientific, 249662) for incubation with anti-PIWI-1 or normal rabbit IgG (Jackson ImmunoResearch, 011-000-003) at 0.2 μg/ml in perm/block (O/N, 4 °C). Samples were washed twice in perm/block before incubation with goat anti-rabbit-Alexa 488 (Jackson ImmunoResearch 111-545-144) at 1:8000 for 30 min at RT. Cells were washed once with perm/block before incubation with DAPI in perm/block at 3 μg/ml for 10 min at RT. Split samples were then resuspended in half of the volume of 1X PBS with 0.5% BSA, filtered through a 40 μm filter (Falcon 352235), and recombined into flow tubes before flow cytometry as above. After gating for singlet and DAPI positive cells, PIWI-LO and PIWI-HI cell fractions were distinguished against side scatter signal (Supplementary Fig. 11d–f).

Data were analyzed and plots were generated in FlowJo (v10.7.1). For irradiation, uninjured planarians were dosed with 60 Grays (6000 rads) using RS-2000 Biological Research X-Ray Irradiator (Rad Source, Buford, GA).

**Cross-referencing of apob(RNAi) RNA-Seq data with published transcriptomes.** For comparison of dysregulated transcripts in apob-M and apob-S animals with bulk neoblast transcriptome data[50] "GSE107874", we first identified "signature" transcripts as follows. "X1 signature" transcripts were defined as those with a log2FC > 0 (FDR < 0.05) compared to both "X2" and "Xins". "X2 signature" transcripts had log2FC > 0 (fdr < 0.05) vs. both "X1" and "Xins". "Xins signature" transcripts had log2FC > 0 (fdr < 0.05) vs. both "X1" and "X2". Similarly, "PIWI-HI signature" transcripts were defined as those with a log2FC > 0 (fdr < 0.05) compared to both "PIWI-LO" and "PIWI-NEG." "PIWI-LO signature" transcripts had log2FC > 0 (fdr < 0.05) vs. both "PIWI-HI" and "PIWI-NEG." "PIWI-NEG signature" transcripts had log2FC > 0 (fdr < 0.05) vs. both "PIWI-HI" and "PIWI-LO." Next, we used the "merge" function and "VennDiagram" package in RStudio (v1.2.1335) to identify signature transcripts in X1/X2/Xins or PIWI-HI/-LO/-NEG

that were also dysregulated (up or down) in apob-M or apob-S animals, as shown in Supplementary Fig. 6. "% of dysregulated transcripts" was calculated as the number of overlapping transcripts divided by the total number of X1/X2/Xins or PIWI-HI/-LO/-NEG transcripts.

For comparison of dysregulated transcripts in apob-M and apob-S animals with single cell type/state data in ref. [40], we again used the "merge" function in RStudio to identify the number of transcripts enriched in individual lineage subclusters (Supplementary Table S2)[40] that were also dysregulated by apob RNAi. "% of dysregulated transcripts" was calculated as the number of overlapping transcripts divided by the total number of transcripts in each individual subcluster. In Fig. 5, the "N/TS" (Neoblast/Transition State) designation included subclusters with high piwi-1 mRNA expression thought to be neoblast/progenitor subpopulations in epidermal, intestine, and protonephridia lineages based on the conclusions of Fincher et al. and other published data[45,66,129–131]. Similarly, the "P" (Progeny) and "M" (Mature) designations were based on conclusions from both single cell RNA-Seq data and previous work. For lineages that are less well understood in vivo (Supplementary Figs. 8 and 9), we designated subclusters/states using both transcript dysregulation in 24 h irradiated animals[50] and piwi-1 mRNA levels in t-SNE plots[40]. "N/TS" subclusters possessed the greatest number of irradiation-dysregulated transcripts and the highest piwi-1 expression; "P" subclusters possessed fewer (by proportion) irradiation-dysregulated transcripts and lower piwi-1 expression; and "M" subclusters had the fewest radiation-sensitive transcripts and negligible piwi-1 expression.

**Image collection and quantification.** Epifluorescent images (FISH samples and cryosections) were collected on a Zeiss AxioObserver.Z1 with Excelitas X-Cite 120 LED Boost illumination and Zen 2.3 Blue (v2.3.64.0). For quantification of pharynx and brain size, organ area and animal area were measured in ImageJ[132] and organ-to-body size ratios were calculated. For intestine, length of anterior branch and posterior branches were measured in ImageJ. Means of posterior branch length were calculated, and then posterior-to-anterior (head fragments) or anterior-to-posterior (tail fragments) length ratio was calculated. For tail fragments with split anterior branch, anterior branch length was measured from the anterior of the pharynx to the tip of the anterior-most primary branch. For tail fragments, the split anterior branch phenotype (failure to fuse at the midline) was scored if there was an obvious gap between anterior branches for at least half the length of the anterior branch. For images of apob-1 and apob-2 FISH on cryosections, z-stacks were collected with an Apotome.2 for generation of maximum orthogonal projections.

For anti-pH3-PS10-labeled samples, z-stacks were collected on a Zeiss AxioObserver.Z1 at 10× magnification, followed by tile stitching, extended depth of focus projection, and background subtraction (PS10 channel only) with a radius of 30. Control and experimental samples to be compared were imaged at identical exposures. For quantification, animal area and PS10+ nuclei were quantified using the Automated Segmentation tools in Zen 2.3 Blue. Briefly, animal area was measured with Gaussian smoothing, no background subtraction or sharpening, Morphology separation, and custom threshold settings that were the same for all samples to be directly compared. PS10+ nuclei number was measured using Lowpass smoothing, Rolling Ball background subtraction, Delineate sharpening, Watersheds separation, with threshold settings and other parameters that were identical for all samples to be directly compared. Number of mitoses per area were calculated for each animal/fragment.

Confocal images were collected on a Zeiss LSM 710 (ZEN version 11.0.3.190, 2012-SP2) or LSM 880 (Zen Black version 2.3 SP1 FP3) laser scanning microscope with 10× Plan NeoFluar, 20X Plan Apo, or 40X C-Apo objectives. For orthogonal projections (anti-ApoB1 immunolabeling and ldlr/vldlr FISH), between three and ten z-planes were collected at 1–2X "optimal" section thickness (based on objective NA). For notum and wnt11-2 FISH, full fragment thickness stacks were projected to ensure that all mRNA-positive cells were counted.

Images of Oil-Red-O-stained sections, live animals, live regenerates, and WISH samples were collected on a Zeiss Stemi 508 with an Axiocam 105 color camera (ZEN 2.3 Blue), or a Zeiss Axio Zoom.V16 with an Axiocam 105 camera (ZEN 2012 Blue). In some cases, brightness and/or contrast were adjusted in Adobe Photoshop to improve signal contrast for Figure panels. Original fluorescent intensity was always used for automated quantification (below).

For automated analysis of prog-1 FISH and anti-PIWI-1 colabeling on sections, z-stacks (5 slices at 1 μm interval) were collected on a Zeiss AxioObserver.Z1 with Apotome.2, using a 20× objective, N.A. = 0.8, followed by tile stitching and extended depth of focus projection. Control and experimental samples were imaged at identical exposures. For automated quantification in FIJI (version 2.3.0), DAPI and prog-1 channels were first merged, then Auto Local Threshold (Phansalkar algorithm) was used to generate a cell region mask, followed by Dilation and Watershedding to segment regions. Analyze Particles was used to limit to 4–40 μm diameter regions, and the resulting mask was used to quantify DAPI, prog-1, and PIWI-1 mean fluorescence intensity per region. Cutoff values for DAPI and prog-1 were set using unlabeled samples to exclude false positives due to background noise. Low and high cutoff values for PIWI-1 signal were set to distinguish PIWI-HI, PIWI-LO, and PIWI-NEG cells, using cell proportions from flow cytometry of anti-PIWI-1-labeled cells as a guide. Data generated by FIJI were compiled in R (version 4.1.2).

For ventral epidermal nuclei counting, single epidermal planes were imaged with a Zeiss ApoTome.2 using a 20× objective, N.A. = 0.8. Four to seven regions (50–100 μm$^2$) in the anterior third of the prepharyngeal region were delineated in FIJI, avoiding regions with gaps caused by N-Acetyl-L-Cysteine treatment. Nuclei were counted using the "FindMaxima" function with Prominence > 300. Nuclear density was calculated per animal by adding total number of nuclei and dividing by total area.

**Statistics.** Detailed data and information regarding statistical testing, n values, and replicate information are included in Source Data, Reporting Summary, and Figure Legends. For experiments with statistical analysis, n values are indicated exactly by the number of data points in figure plots, along with definitions of error bars and p or q values in Legends. All tests were conducted in Prism 9 (GraphPad Software, San Diego, CA). For one-way ANOVA, ordinary ANOVA was performed unless Brown–Forsythe and Bartlett's tests indicated standard deviations were significantly different, then Brown–Forsythe and Welch ANOVA tests were performed. For two-way ANOVA (flow cytometry experiments with irradiation), q values (FDR-adjusted p values) were reported when interaction between RNAi condition ("Genotype") and irradiation was significant (X2 subpopulation in Fig. 4i). Otherwise, p values were derived using one-way ANOVA followed by Tukey's or Dunnett's multiple comparisons test. p values of <0.05 (*), <0.01 (**), <0.001 (***), and <0.0001 (****) were annotated with asterisks in figures.

For statistical testing of overlap between genes dysregulated in apob(RNAi) planarians and other RNA-Seq data sets, the R Package GeneOverlap[133] was used to conduct Fisher's exact test on each comparison. Total number of detected transcripts ("genome size" in GeneOverlap v1.20.0) was determined conservatively by only including transcripts detected in both apob RNA-Seq and bulk[50] or single cell[40] sequencing data. For sc-RNA-Seq, the digital expression matrix in GEO "GSE111764" was normalized in Seurat as in Fincher et al.[40]; only transcripts with non-zero expression in 0.5% of cells (Fincher TableS1) in each subcluster were considered to be detected. p values <0.05 are indicated with a caret (^) in figures, and provided individually in Source Data.

Replicate and statistical information for RNA-Seq and other bioinformatics experiments are detailed in appropriate Methods sections.

**Reporting summary.** Further information on research design is available in the Nature Research Reporting Summary linked to this article.

## Data availability
Raw and processed RNA-Seq data associated with this study are available in the NCBI Gene Expression Omnibus (GEO) under records "GSE174246", "GSE174227", and "GSE174228". Other data supporting this study's findings are available within the article and its Supplementary files, or from the authors upon reasonable request. Source data are provided with this paper.

## Code availability
FIJI macro and R scripts related to cross-referencing of bulk and single cell RNA-Seq data and cell quantification on tissue sections are available at Zenodo.org "zenodo.6596520" and "zenodo.6596518".

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

## Acknowledgements

We are very grateful to Phil Newmark (HHMI and the Morgridge Institute for Research), in whose laboratory this project was initiated. We thank members of the Newmark and Forsthoefel labs, and OMRF colleagues Pat Gaffney, David Jones, Linda Thompson, Dean Dawson, and Hui-Ying Lim for insightful discussions. We thank Jochen Rink, James Cleland, and Hanh Vu (Max Planck Institute for Biophysical Chemistry) for sharing planarian protein extraction protocols, Vasileios Morkotinis for *tgs-1* plasmid, and Rachel Roberts-Galbraith (Univ. of Georgia) for *notum* and *wnt11-2* plasmids. We thank Steve Farber (Carnegie Institution for Science) for advice on lipid extraction and TLC, and Jayhun Lee (The University of Texas Health Science Center at Houston) for collaborative development of planarian lipid analysis protocols. We are grateful to Hadi Maktabi (ProteinSimple) and to OMRF colleagues Summer Wang and Lin Wang for help in developing Wes protocols; and to members of the OMRF Flow Cytometry Core (Jacob Bass and Diana Hamilton), the Quantitative Analysis Core (Lori Garman and Nathan Pezant), the Imaging and Histology Core, the Clinical Genomics Core, the Gnotobiotic Mouse Core (for use of X-irradiator), the Center for Biomedical Data Science, and IT/Research Computing Services for invaluable technical assistance. JRO was supported by the Summer Research Opportunities Program at the University of Illinois at Urbana-Champaign. This work was supported by NIH Centers of Biomedical Research Excellence (COBRE) GM103636 (Project 1 to D.J.F.), and the Oklahoma Medical Research Foundation.

## Author contributions

Conception and design of the project: D.J.F. and L.L.W.; data collection: L.L.W., C.G.B., J.R.O., N.I.C., M.R.D., and D.J.F.; data analysis and visualization: L.L.W., C.G.B., N.I.C., and D.J.F.; R and ImageJ scripts: D.J.F., N.I.C., and L.L.W.; data interpretation: L.L.W., C.G.B., and D.J.F.; RNA-seq analysis: D.J.F.; manuscript preparation: D.J.F., L.L.W., and C.G.B.

## Competing interests

The authors declare no competing interests.
