## [Peer Review File · Nature Communications]

REVIEWER COMMENTS

Reviewer #1 (Remarks to the Author):

Lipid metabolism plays an instructive role in regulating stem cell state and differentiation. However, the roles of lipid mobilization and utilization in stem cell-driven regeneration are unclear. Planarian flatworms readily restore missing tissue due to injury-induced activation of pluripotent somatic stem cells called neoblasts. In this study, the authors identify two intestine-enriched orthologs of apolipoprotein b, apob-1 and apob-2. They found: (a) ApoB is expressed by phagocytes in the intestine, a primary site of LP production and secretion; (b) ApoB mediates secretion of neutral lipids in LPs from intestinal phagocytes to neoblasts and their progeny; (c) In the absence of ApoB, lipids accumulate in the intestine, and LP delivery to neoblasts and their progeny is disrupted, reducing their neutral lipid content. Neoblast proliferation and renewal are largely unaffected by reduced ApoB function. Instead, differentiation and later maturation of most, if not all, planarian cell lineages are slowed, causing an accumulation of differentiating progeny and a delay in regeneration of multiple organs.

Then, they did gene profile and found that apob RNAi causes expansion of the population of differentiating neoblast progeny and dysregulates expression of genes enriched in differentiating and mature cells in eight major cell type lineages. They conclude that intestine-derived lipids serve as a source of metabolites required for neoblast differentiation.

In summary, this is just a phenotypic description and lacks detail mechanism. Then, in discussion they made a number of guesses about mechanisms:

- (1). ApoB inhibition could dysregulate epigenetic changes through multiple pathways.**
- (2). ApoB depletion may also delay differentiation by other mechanisms.**
- (3). Additionally, LP-mediated transport of morphogens like Hedgehog or Wnt proteins, whose planarian orthologs play important roles in regulating axial polarity and tissue differentiation, may be affected by apob knockdown.**
- (4). apob RNAi causes moderate downregulation of most fibrillar collagens, as well as the basement membrane collagen4-1, which promotes differentiation (Supplementary Data 4 and 5). Thus, ApoB depletion may also delay differentiation indirectly, by compromising the generation and/or function of muscle cells.**

In my view, this is not ready to be published on a major Journal.

Reviewer #2 (Remarks to the Author):

Wong et al. performed a comprehensive study investigating the role of apolipoprotein b orthologs in planarians. Authors reported that apolipoprotein b orthologs are expressed by intestinal cells and serve as neutral lipid transporter to other tissues. Authors demonstrated that apob RNAi results in the accumulation of neutral lipid and inhibit/delay the regeneration in planarians. RNAi also causes head regression and lysis in intact animals. Further investigations showed that apob RNAi does not affect stem cells proliferation or survival. The authors then performed FASC analysis on cells isolated from apob RNAi animals and observed an increase in the stem cell progeny (X2) population. The transcriptomic analysis also showed the preferential deregulation of transcripts in the stem cell progeny and mature cells. Hence authors finally concluded that apolipoprotein b orthologs regulate the differentiation

of stem cell progeny in the planarians.

Overall, this work is novel, exciting and well-executed. The methodology used is sound and has provided enough details. This study would be of significant interest to the planarian community. I recommend publication with the addition of few experiments that will provide direct evidence to the authors' conclusions.

Experimental suggestions:

1. Fig2C and D: Please mention the observation time point post-RNAi. Also, the NL distribution in Fig2D should be shown along with the marker for intestinal cells to better understand NL's spread in the tissue in both control and RNAi condition.

2. Line number 177-179: The accumulation of neutral lipids at day 6 in control (SuppFig2F) is apparently decreased compared to day 3 control. It is also reduced in apob(RNAi) animals at day 6 compared to day 3.

This decrease raises several interesting questions like -

(i) What happens at later time points during regeneration? Does it get increased again even without feeding animals?

(ii) If yes, then what could be the source of those neutral lipids?

(iii) If no, then how does this reduced amount of neutral lipids support regeneration during the second round of amputation. (As far as I know, planarians regenerate successfully even after repetitive rounds of amputation without feeding)

As this is the first paper showing the accumulation of neutral lipids in planarians, it would be nice to see detailed kinetics to neutral lipids during regeneration (from day 1 to day 9/more on every day or every alternate day) in both control and apob(RNAi) animals. This will give a clear picture of the dynamics of neutral lipids over the entire period of regeneration.

3. Line number 182: The stainings showing up-regulation of lipoprotein receptors are not consistent between different images.

The FISH staining for ldlr-1 and ldlr-2 in SuppFig3d doesn't look very specific. Staining looks homogenous throughout the animal without any obvious overexpression in the blastema region, as shown in SuppFig3b.

In the case of vldlr-1 the staining in the existing pharynx is not apparent in the SuppFig3b, but it's evident in the SuppFig3d.

4. It feels like the intestinal branches in apob(RNAi) animals are broader in Fig3G, Fig3H. Also, Fig5H shows massive up-regulation of mature intestinal cell genes.

If we combine both data, we can speculate that there is possibly increased/preferential differentiation of intestinal lineage. Authors should test this by the BrdU or EdU incorporation assay.

If this is true, then it will change the interpretation of other results as well. In that case, we may argue that increased NL in apob(RNAi) animals (Fig2D, SuppFig2E) is (partly/entirely) due to increased/preferential differentiation into the intestinal lineage.

5. Several observations (delay in regeneration, increase in X2 population, greater modulation in X2 transcripts) suggest the differentiation defect in stem cell progeny after apob RNAi. However, the authors have not performed any direct experiment to prove the same.

Authors can easily provide direct quantitative evidence of accumulation of stem cell progeny after apob(RNAi) by looking at any well-characterized lineage in the planarians (e.g. epidermal). Also, BrdU or EdU incorporation studies will provide direct evidence lack of differentiation of X2 cells into Xins cells.

Minor Comments:

1. As data demonstrates that apolipoprotein b orthologs are required for differentiation of "stem cell progeny to mature cell" and not for differentiation of "stem cell to progeny", several statements (mentioned below) in the manuscripts needs to be modified appropriately to avoid possible confusion.

Title: Instead of saying "...stem cell differentiation", it would be more appropriate to say "...stem cell progeny differentiation"

Line number 36: Please mention "neoblast progeny differentiation" instead of "neoblast differentiation"

Line number 100: Please mention "stem cell progeny differentiation" instead of "stem cell differentiation"

Line number 358: "ApoB was required for differentiation of planarian neoblasts" should be "ApoB was required for differentiation of planarian neoblast progeny"

Line number 361: "...planarian neoblast differentiation" should be "...planarian neoblast progeny differentiation"

Line number 363: "...differentiation of neoblast is delayed" should be "...differentiation of neoblast progeny is delayed"

2. Define "apob-M" and "apob-S" at it's first use. These abbreviations are first used in Fig2I and J but not defined there.

**3. Fig2F: Does Xins contains "fully differentiated/mature" cells of "late/mature progeny" as mentioned in the figure, or it's a mixture of both?
Also, please label "S/G2/M" cell population as "X1".**

4. Line number 170-173: I don't see an apparent up-regulation at neoblast proliferation (1-3 days) time points in SuppFig2A and B. The graphs will be clearer if the authors mention asterisks at places where there is a significant fold change.

5. Line number 175-176: Is the increase at Day 3 significant? Again, mentioning asterisk at places where there is a significant fold change will help the reader.

6. SuppFig4B: It would be more appropriate to say "pluripotent neoblasts" rather than "totipotent neoblasts".

7. Line number 399-403: The sentence needs to be rephrased as this is just speculation, and there is no direct evidence.

Reviewer #3 (Remarks to the Author):

Summary: An extensive literature demonstrates the importance of lipid metabolism in regulating stem cell biology. However, the role of lipid metabolism during regeneration, especially whole body regeneration, remains largely unexplored. In this study, Wong and co-authors explore the role of lipid metabolism in neoblast function during planarian regeneration. Previous work from this group demonstrated that knocking down a

transcription factor (Nkx2.2) expressed in the intestine inhibited the formation of the blastema during regeneration. This intriguing result suggested that the intestine is playing an unknown role in regulating neoblasts. To identify possible mechanisms, the authors performed RNA-seq and found that two orthologs of apolipoprotein b (apob-1 and apob-2) is highly down regulated in Nkx2.2 knockdown animals. Apolipoproteins mediate trafficking of NLs to LPs from digestive/storage organs to other tissues, thus suggesting a role for apoproteins in providing neoblasts with lipid during regeneration. Knocking down both apob isoforms led to a range of phenotypes, including head regression – this is indicative of a neoblast defect. The authors made a planarian specific antibody for ApoB-1 protein and found that although the apob mRNA is found almost entirely in the intestine, the ApoB-1 protein is found outside of the intestine, which is consistent with a conserved function for ApoB in lipid transport. Furthermore, in apob RNAi animals, the neoblasts had a lower NL content, suggesting that ApoB functions to transport lipids from the intestine to neoblasts. During regeneration, the apob genes as well as other genes involved in lipid metabolism are upregulated and apob RNAi led to regeneration delays and defects, which are likely explained by a delay or defect in neoblast differentiation. Accordingly, RNA-seq of apob RNAi animals demonstrated that differentially expressed genes were enriched for transcripts found in differentiating neoblast progeny and all lineages were affected. The authors speculate that lipids provided to neoblasts by ApoB could be a carbon source for the epigenetic modifications that are required for large scale neoblast differentiation and/or the lipids could be used as an energy source.

This study is carefully done, well-written, and well presented. The authors exploration of lipid metabolism during planarian regeneration is novel and important. I have only a few very minor comments:

1. Was the antibody designed to only recognize ApoB-1 specifically, or is it expected to cross react with ApoB-2? This should be better explained, at least in the methods.
2. The meaning of ApoB-S and ApoB-M should be explained in the figure legends

To all Reviewers

We sincerely thank all three reviewers for their positive comments and constructive critiques. Addressing their concerns, here and in the revised text, has significantly improved the manuscript, especially with regard to more rigorous testing of the idea that ApoB is required for differentiation of planarian stem cell progeny. To address this concern, we developed new methods using the PIWI-1 marker to quantify accumulating progeny. We include two additional figures (Fig. 6 and Supplementary Fig. 9) in the revised manuscript, and two figures for reviewers and a list of edits to figures and supplementary data at the end of this response letter. We also thank Reviewer 1 for the opportunity to make clear the novel and significant aspects of our study that will impact multiple fields, warranting publication in *Nature Communications*. We address each reviewer's comments in detail below.

(Please note that line references refer to the PDF version with tracked changes.)

Reviewer 1 (Remarks to Authors)

Lipid metabolism plays an instructive role in regulating stem cell state and differentiation. However, the roles of lipid mobilization and utilization in stem cell-driven regeneration are unclear. Planarian flatworms readily restore missing tissue due to injury-induced activation of pluripotent somatic stem cells called neoblasts. In this study, the authors identify two intestine-enriched orthologs of apolipoprotein b, apob-1 and apob-2. They found: (a) ApoB is expressed by phagocytes in the intestine, a primary site of LP production and secretion; (b) ApoB mediates secretion of neutral lipids in LPs from intestinal phagocytes to neoblasts and their progeny; (c) In the absence of ApoB, lipids accumulate in the intestine, and LP delivery to neoblasts and their progeny is disrupted, reducing their neutral lipid content. Neoblast proliferation and renewal are largely unaffected by reduced ApoB function. Instead, differentiation and later maturation of most, if not all, planarian cell lineages are slowed, causing an accumulation of differentiating progeny and a delay in regeneration of multiple organs.

Then, they did gene profile and found that *apob* RNAi causes expansion of the population of differentiating neoblast progeny and dysregulates expression of genes enriched in differentiating and mature cells in eight major cell type lineages. They conclude that intestine-derived lipids serve as a source of metabolites required for neoblast differentiation.

In summary, this is just a phenotypic description and lacks detail mechanism. Then, in discussion they made a number of guesses about mechanisms:

- (1) ApoB inhibition could dysregulate epigenetic changes through multiple pathways.**
- (2) ApoB depletion may also delay differentiation by other mechanisms.**

(3) Additionally, LP-mediated transport of morphogens like Hedgehog or Wnt proteins, whose planarian orthologs play important roles in regulating axial polarity and tissue differentiation, may be affected by apob knockdown.

(4) *apob* RNAi causes moderate downregulation of most fibrillar collagens, as well as the basement membrane *collagen4-1*, which promotes differentiation (Supplementary Data 4 and 5). Thus, ApoB depletion may also delay differentiation indirectly, by compromising the generation and/or function of muscle cells.

In my view, this is not ready to be published on a major Journal.

Authors' Response to Reviewer 1

We are pleased that the reviewer agreed with the importance of understanding lipid mobilization and utilization by stem cells during regeneration, and that they had no criticism of our data or conclusions that ApoB-mediated lipid transport and lipid metabolism are required for stem cell progeny differentiation and regeneration. However, the reviewer felt the study needed to determine the detailed mechanism with respect to how ApoB and intestine-derived lipids are required for neoblast differentiation to warrant publication.

We agree with the reviewer that identifying the mechanism(s) by which *apob* and intestine-derived lipids influence stem cell differentiation will be important. Nonetheless, we feel that our study is, in fact, mechanistic. We use multiple methods and rigorous analysis (including additional approaches in the revised manuscript) to identify the *cellular* mechanism (delayed differentiation) underlying delayed regeneration in *apob* knockdown animals. We also demonstrate using biochemical, histological, and flow cytometric methods that the *molecular* function of ApoB orthologs as secreted regulators of lipid transport has been conserved in planarians. We estimate that testing the multiple hypotheses we raise in the Discussion about how reduced lipid secretion affects neoblast differentiation would require 2-3 years of additional effort to address at a level of depth and thoroughness appropriate for *Nature Communications*. Given that the revised manuscript now has seven figures and 10 supplemental figures, we also believe that additional work would make the manuscript excessively long for this journal's format.

We also suggest that such additional studies are not required for our manuscript to have a significant impact on multiple fields. Impact is derived not only from detailed molecular analysis of an entire biological process, but also the rigor of the study, and the significance and novelty of the findings. Examples of such manuscripts in *Nature Communications* include "Regenerative capacity in newts is not altered by repeated regeneration and ageing" (<https://doi.org/10.1038/ncomms1389>), "Parasympathetic stimulation improves epithelial organ regeneration" (<https://doi.org/10.1038/ncomms2493>), and "A stem cell population at the anorectal junction maintains homeostasis and participates in tissue regeneration," (<https://doi.org/10.1038/s41467-021-23034-x>). We note that our study is the first to

definitively identify a functional role for planarian intestinal lipid stores in regeneration, despite over a century of study at more superficial histological and biochemical levels. Additionally, to our knowledge, this is also the first study to uncover a functional role for lipoproteins in an emerging regeneration model, and one of only a handful (in any system) to reveal a direct link between lipoprotein trafficking and stem cell regulation. Finally, cross-referencing of our gene expression data with published single cell state data, and utilization of anti-PIWI-1 labeling by both flow cytometry and on tissue sections (added to the revised manuscript, please see below) to assess planarian phenotypes quantitatively are both novel applications of previous discoveries by others, and will be of specific interest to the planarian research community.

Again, we agree with the reviewer that mechanism will be important to investigate further, and we are very interested in investigating the possibilities we raise in the Discussion. However, we believe our rigorous molecular and cellular analysis, the development of new approaches and reagents, and the novelty of our findings will all be of wide interest to researchers studying planarians, regeneration, and lipid metabolism, and that the study makes a high impact contribution that is appropriate for the *Nature Communications* readership.

Reviewer 2 (Remarks to Authors)

Wong et al. performed a comprehensive study investigating the role of *apolipoprotein b* orthologs in planarians. Authors reported that *apolipoprotein b* orthologs are expressed by intestinal cells and serve as neutral lipid transporter to other tissues. Authors demonstrated that *apob* RNAi results in the accumulation of neutral lipid and inhibit/delay the regeneration in planarians. RNAi also causes head regression and lysis in intact animals.

Further investigations showed that *apob* RNAi does not affect stem cells proliferation or survival. The authors then performed FASC analysis on cells isolated from *apob* RNAi animals and observed an increase in the stem cell progeny (X2) population. The transcriptomic analysis also showed the preferential deregulation of transcripts in the stem cell progeny and mature cells.

Hence authors finally concluded that *apolipoprotein b* orthologs regulate the differentiation of stem cell progeny in the planarians.

Overall, this work is novel, exciting and well-executed. The methodology used is sound and has provided enough details. This study would be of significant interest to the planarian community. I recommend publication with the addition of few experiments that will provide direct evidence to the authors' conclusions.

Authors' Response to Reviewer 2

We are grateful to the reviewer for their positive comments and constructive suggestions for additional experiments, and for numerous helpful suggestions to clarify and improve the text, figure legends, and figures.

In particular, addressing experimental suggestion 5, to provide "direct evidence to the author's conclusions," has significantly strengthened the manuscript. We addressed this point by generating a PIWI-1 antibody, and then using this antibody to directly quantify neoblast progeny in the epidermal lineage (as the reviewer suggested), as well as globally. These analyses required us to develop new methods for automated image analysis, and to adapt previously published methods for flow cytometric quantification of fixed planarian cells. Together, the results of these additional experiments (described in detail below) demonstrate that cells expressing low levels of the PIWI-1 protein (marking their state as early progeny) accumulate in *apob* RNAi animals, providing direct evidence that ApoB is required for differentiation. In addition, these experiments also enabled discovery of evidence of a moderate delay in the cell cycle, which was not observed in the *in vivo* experiments in Supplementary Fig. 4, or in original live cell flow cytometry experiments (Fig. 4 and Supplementary Fig. 4). We have also addressed the reviewer's concern regarding the intestinal lineage using EdU pulse-chase labeling and other methods, and we have addressed all concerns regarding the text, figures, and legends by editing as the reviewer suggested.

Reporting these additional results has required the inclusion of an additional figure (Fig. 6) and supplementary figure (Supp. Fig. 9), additional Results (lines 393-450), edits to the Discussion (lines 456-459) and accompanying Methods (lines 593, 616, 635, 647-649, 652-666, 699-710, 844-867, 934-954, 966, and 971-972).

Again, we thank the reviewer for their rigorous review and thoughtful suggestions. Addressing most of the reviewer's recommendations (below) has significantly strengthened the manuscript.

Experimental suggestions:

1. Fig 2c and d: Please mention the observation time point post-RNAi. Also, the NL distribution in Fig 2d should be shown along with the marker for intestinal cells to better understand NL's spread in the tissue in both control and RNAi condition.

We agree with the reviewer that distinguishing intestinal tissue from surrounding tissue is informative. Qualitatively, in uninjured animals (Fig. 2d) and regenerates (Supp. Fig. 2e and f), there is an accumulation of neutral lipids outside the intestine. This is consistent with the roles of ApoB proteins in receptor-mediated uptake/metabolism (as well as NL secretion), and with our demonstration that NL content is lower in neoblasts and their immediate progeny (Figs. 2i-j).

Unfortunately, processing for WISH/FISH/immunofluorescence with intestinal markers makes it difficult or impossible to simultaneously label intestinal tissue along with lipids. This is due to the use of solvents and detergents (which extract lipids) required for labeling. However, the gut can be distinguished morphologically using the subtle light scattering differences by this tissue in histological sections. We have added dotted outlines in Fig. 2d, and Supp. Fig. 2e and f, to enable readers to distinguish NLs outside the intestine.

As the reviewer suggested, we have also indicated the time point (7 days after the last dsRNA feeding) for Fig. 2c-d, and for Supp. Fig. 2e-f (animals were amputated 7 days after the last dsRNA feeding) in the figure legends. In all experiments except the initial viability testing in Fig. 2a, we fixed, amputated, or dissociated animals (for flow cytometry) 7 days after the last dsRNA feeding. We have added this information to the Methods (lines 647-649). We thank the reviewer for bringing this oversight to our attention.

2. Line number 177-179: The accumulation of neutral lipids at day 6 in control (Supp Fig 2f) is apparently decreased compared to day 3 control. It is also reduced in *apob(RNAi)* animals at day 6 compared to day 3. This decrease raises several interesting questions like –

(i) What happens at later time points during regeneration? Does it get increased again even without feeding animals?

(ii) If yes, then what could be the source of those neutral lipids?

(iii) If no, then how does this reduced amount of neutral lipids support regeneration during the second round of amputation. (As far as I know, planarians regenerate successfully even after repetitive rounds of amputation without feeding.)

As this is the first paper showing the accumulation of neutral lipids in planarians, it would be nice to see detailed kinetics to neutral lipids during regeneration (from day 1 to day 9/more on every day or every alternate day) in both control and *apob(RNAi)* animals. This will give a clear picture of the dynamics of neutral lipids over the entire period of regeneration.

We agree that what happens to intestinal lipid content at later time points or after multiple rounds of regeneration, molecular regulation of lipid synthesis, and detailed kinetics of lipid content during regeneration would all be fascinating and logical next steps suggested by our observations. However, the results of such additional experiments would not strengthen or alter our conclusions, but would extend our study into new areas that we respectfully suggest are beyond the scope of the current manuscript, and would be more appropriate for a follow-up project. For example, a more detailed analysis of neutral lipid content in the intestine would likely require multiple quantitative approaches (histological and biochemical) to be informative, and might also require identification of lower dsRNA doses to minimize the eventual lethality of *apob* knockdown (Fig. 2a), potentially confounding analysis. In addition, identification of sources of new lipids after regeneration is likely to require molecular analysis/screening of a large array of additional regulators of neutral lipid biogenesis, including mono- and diglyceride acyltransferases, acyl-CoA cholesterol acyltransferases, and upstream *de novo* fatty acid

synthetases (which might not even have orthologs in planarians, please see Grohme et al., *Nature*, 2018, <https://doi.org/10.1038/nature25473>). This would require knockdowns, and also quantitative histological and biochemical methods to identify whether lipid biogenesis was affected, and whether this impinged upon regeneration. We thank the reviewer for this interesting suggestion, but we believe that rigorous efforts to address these questions would likely constitute a substantial follow-up manuscript.

We agree that such experiments will be important future directions. Therefore, we have incorporated the point about what happens at later time points as a potential future direction in the Discussion, when we consider how lipid regulators' expression is coordinated. We provide additional support for the possibility that planarians restore NLs or adapt to reduced NL levels by pointing out that some lipid-related genes are up- and down-regulated at later regeneration time points (Supplementary Fig. 3e) (lines 556-562):

Furthermore, some lipid regulators are up- and down-regulated at later regeneration time points (Supplementary Fig. 3e). This raises the intriguing possibility that planarians may replenish NLs utilized during regeneration (Supplementary Fig. 2e, f), or adapt metabolic networks to reduced lipid levels, especially in the absence of post-regeneration feeding. Identification of additional lipid regulators required at different stages of regeneration, and unraveling which transcription factors, chromatin modifiers, and other factors control their expression are thus additional priorities for future study.

3. Line number 182: The stainings showing up-regulation of lipoprotein receptors are not consistent between different images. The FISH staining for *ldlr-1* and *ldlr-2* in Supp Fig 3d doesn't look very specific. Staining looks homogenous throughout the animal without any obvious overexpression in the blastema region, as shown in Supp Fig 3b. In the case of *vldlr-1* the staining in the existing pharynx is not apparent in the Supp Fig 3b, but it's evident in the Supp Fig 3d.

We agree with the reviewer that there were apparent inconsistencies in labeling in this figure, and we thank the reviewer for the opportunity to clarify the reasons for this in the figure legend.

ldlr-1, *ldlr-2*, and *vldlr-1* do have broad expression patterns. This is also supported by single cell RNA-Seq data in regenerates: we provide single cell RNA-Seq UMAP plots from Benham-Pyle et al., *Nat. Cell Biol.*, 2021 (<https://doi.org/10.1038/s41556-021-00734-6>) at the end of this response letter in support of our FISH results (Reviewer Fig. 1).

The apparent discrepancy between the WISH images in Supp. Fig. 3b and the FISH images in Supp. Fig. 3d is likely due to underdevelopment of colorimetric signal in Supp. Fig. 3b samples. In colorimetric WISH, signal development occurs over 1-2 hours. In the blastema, signal sometimes develops more quickly than in pre-existing tissues due to the greater sensitivity of blastema tissue to proteinase K digestion (King and Newmark, *BMC Dev. Biol.*, 2013, <https://doi.org/10.1186/1471-213X-13-8>), and more rapid permeation of substrate in this region. Tyramide-based FISH reactions, by contrast, are usually complete within 10-15 minutes

of development (King and Newmark, *BMC Dev. Biol.*, 2013), explaining why these samples have higher signal in pre-existing (distal to the blastema) tissue, as compared to WISH samples. We stopped development of the WISH in situ early, to show expression in new tissue, but we agree that the discrepancy between the WISH and FISH images is misleading.

To address this, we have added a note in the Supp. Fig. 3d legend explaining the differences:

Expression in pre-existing tissue relative to the blastema is higher in FISH samples as compared to WISH samples (b), likely due to more rapid completion of the tyramide signal amplification reaction relative to colorimetric development.

In addition, we consistently observe upregulation in regenerating pharynx (e.g., 5 dpa heads in Supp. Fig. 3b and 5 dpa tails in Supp. Fig. 3d), but not in uninjured animals or trunk fragments (e.g., 5 dpa trunks in Supp. Fig. 3b), suggesting that *ldlr-1*, *ldlr-2*, and *vldr-1* expression is eventually downregulated in the pharynx. We have also added a brief note mentioning this in the Supp. Fig. 3b legend:

Expression is lower in pharynges in uninjured animals and in the pre-existing pharynx in trunk fragments, suggesting downregulation when regeneration is complete.

4. It feels like the intestinal branches in *apob(RNAi)* animals are broader in Fig 3g, Fig 3h. Also, Fig 5h shows massive up-regulation of mature intestinal cell genes. If we combine both data, we can speculate that there is possibly increased/preferential differentiation of intestinal lineage. Authors should test this by the BrdU or EdU incorporation assay. If this is true, then it will change the interpretation of other results as well. In that case, we may argue that increased NL in *apob(RNAi)* animals (Fig 2d, Supp Fig 2e) is (partly/entirely) due to increased/preferential differentiation into the intestinal lineage.

The reviewer raises an insightful and testable interpretation, that increased width of intestinal branches and elevated gene expression in the intestinal lineage could be explained by increased or even preferential differentiation of intestinal cells. Upon further analysis, however, we find little support for such a hypothesis, and we suggest that additional factors and complexity must also be considered when interpreting the observations the reviewer mentions.

Theoretically, preferential changes in differentiation could be caused by alterations in specification of neoblasts, transdifferentiation from non-intestinal to intestinal lineages after specification, or slower differentiation at later stages of some lineages or faster differentiation of intestinal cell types.

With regard to specification, we do not believe that ApoB inhibition blocks specification globally, and we discuss this in the Discussion, paragraph 2. It is formally possible that a more selective skewing of specification towards intestinal lineages could occur. However, we do not observe a significant reduction in *prog-1+* early progeny cells in the epidermal lineage, which is likely to be one of the most abundant populations of early epidermal progenitors (please see

Plass et al., *Science*, 2018, Fig. S1E, DOI: 10.1126/science.aaq1723). This is demonstrated in Fig. 6c in the revised manuscript; please also see our response to experimental suggestion 5, below. In addition, although gene expression does not necessarily correlate with cell count, we also only observe very modest (1.25X) upregulation of 2 of the 4 best-characterized intestinal progenitor markers: *gata4/5/6-1*, *prox-1*, *hnf-4*, or *nkx2.2* in *apob-S* animals, but not *apob-M* animals (Supplementary Data 4 and below). Thus, the idea that specification is either affected globally or selectively is not supported by our data.

Gene	Transcript	apob-M log ₂ FC	apob-M FDR	apob-S log ₂ FC	apob-S FDR
gata4/5/6-1	dd_Smed_v6_4075_0_1	0.117	0.371924542	0.309	0.000331042
prox-1	dd_Smed_v6_13772_0_1	0.374	0.098988023	0.159	0.53090005
hnf-4	dd_Smed_v6_1694_0_1	0.065	0.695228547	0.157	0.134637506
nkx2.2	dd_Smed_v6_2716_0_1	0.123	0.335180549	0.318	0.000179054

Similarly, transdifferentiation is not known to occur in planarians, and is also not supported by any available single cell data.

To test the third possibility, that there might be increased or faster differentiation in the intestinal vs. other lineages, we used three approaches.

First, as the reviewer suggested, we used EdU pulse-chase analysis of intestinal differentiation in 6 day post-amputation regenerates. We provide a figure with the results (Reviewer Fig. 2) and methods at the end of this response letter.

We did not observe an increase in EdU+ intestinal cells in *apob-M* ("mild") regenerates (Reviewer Fig. 2b). We did observe an increase in *apob-S* ("severe") regenerates, although this was not statistically significant (Reviewer Fig. 2b). The increase in EdU+ intestinal cells in regenerates could be evidence of greater differentiation of cells into the intestine, as the reviewer proposes.

However, if increased intestinal differentiation occurs in *apob(RNAi)* animals, then we would predict a detectable increase in the overall number of intestinal cells. As a second approach, therefore, and to test this prediction, we also calculated the number of DAPI+ cells per unit area (mm²) in both the intestine, as well as in the entire pre-pharyngeal area scored in each tissue section (Reviewer Fig. 2c-e).

In this analysis, we observed no increase in the number of DAPI+ intestinal cells relative to intestinal area, or to total area scored (Reviewer Fig. 2c-e), indicating that overall, the number of intestinal cells in regenerating branches is unchanged, even in *apob-S* animals. This suggested that if there is an increase in intestinal differentiation (particularly in *apob-S* regenerates, in which we observed a modest increase in EdU+ intestinal cells), it might be balanced by cell loss/death, either at later stages of differentiation, or of mature cells, some of which might have existed prior to injury. These results also illustrate that the increased

percentages of upregulated intestine-enriched genes (Fig. 5h) does not necessarily correlate with intestinal cell number, and could have other causes. For example, intestinal cells might mount compensatory gene expression responses to lipid accumulation, a possibility we raised more generally in our original Results (lines 320-323 in the revised manuscript).

Interpretation of EdU pulse-chase experiments is also confounded by the possibility that post-mitotic EdU+ cells might integrate into the intestine very early during their differentiation. Thus, EdU pulse-chase experiments do not necessarily enable detection of later differentiation delays such as progeny accumulation, which we proposed as the primary defect in *apob(RNAi)* animals in our original manuscript. (We have demonstrated this directly in the revised manuscript, please see also our response to experimental suggestion 5, below).

Therefore, as a third measure of whether differentiation was affected, we also labeled these sections with the PIWI-1 antibody we generated. PIWI-1 protein perdures in early neoblast progeny (please see also our response to suggestion 5, below) (Guo et al., *Dev. Cell*, 2006, DOI: 10.1016/j.devcel.2006.06.004). Therefore, increased fractions of cells expressing high ("PIWI-HI") or low ("PIWI-LO") PIWI-1 could indicate delays in differentiation. Although none of the differences between control and *apob(RNAi)* regenerates was significant, we did observe clear upward trends in the percentages of EdU+ intestinal cells (Reviewer Fig. 2f-g) and DAPI+ intestinal cells (Reviewer Fig. 2i-j) that expressed low or high levels of PIWI-1, and downward trends in the percentages of these populations not expressing PIWI-1 ("PIWI-NEG", Reviewer Fig. 2hand 2k). These results suggest that differentiation is delayed after EdU+ progeny in the intestinal lineage have integrated into the intestine proper, and are consistent with our analysis of the epidermal lineage and global progeny quantification in Fig. 6 (discussed in response to the next suggestion).

Furthermore, several other causes might explain the apparent thickness of the intestinal branches in Fig. 3g-h. These could include (a) increased lipid content and therefore size of intestinal phagocytes (please note the distension of the intestine in Oil Red O-labeled samples in Fig. 2 and Supplementary Fig. 2e) or (b) a failure of intestinal branches to elongate during remodeling (paralleling the failure of the fragments to re-scale their length-to-width ratios, most likely caused by delayed re-expression of polarity cues), altering branch diameter.

We thank the reviewer for suggesting this possibility, and we believe it was worthwhile to investigate further. However, we do not think our data support preferential differentiation of intestinal cells. Together, the lack of evidence for altered specification, the unlikelihood of transdifferentiation, the possibility of increased cell death in *apob-S* animals (an interesting future direction), the similar overall numbers of intestinal cells in control and *apob(RNAi)* animals, and the possibility of compensatory gene expression as an explanation for upregulation of intestinal transcripts, all suggest that preferential intestinal differentiation does not occur to a significant degree. Furthermore, the upward trend in PIWI-1+ cells in the intestine strengthens our original conclusion that ApoB inhibition delays differentiation after specification, which we demonstrate directly in response to the next experimental suggestion.

We also thank the reviewer for bringing to our attention that we might not have adequately considered additional interpretations and complexity of our transcriptome data. We do believe that the transcriptome changes caused by ApoB inhibition could have multiple underlying causes, so to address this further, we have added the following text to Results (lines 367-374):

In most subclusters, the proportion of downregulated transcripts was greater, but we also observed upregulated transcripts in nearly all subclusters. In addition, the proportion of upregulated transcripts was greater in some protonephridial, *cathepsin*-positive, pharynx, parenchymal, neural, and most intestine subclusters (Fig. 5c-i, Supplementary Fig. 8c-l). This suggested that ApoB inhibition could cause subtle shifts in the proportions of specific cell types, but also that changes in lipid availability could induce gene expression responses (for example, to compensate for lipid accumulation in the intestine).

5. Several observations (delay in regeneration, increase in X2 population, greater modulation in X2 transcripts) suggest the differentiation defect in stem cell progeny after *apob RNAi*. However, the authors have not performed any direct experiment to prove the same. Authors can easily provide direct quantitative evidence of accumulation of stem cell progeny after *apob(RNAi)* by looking at any well-characterized lineage in the planarians (e.g. epidermal). Also, BrdU or EdU incorporation studies will provide direct evidence lack of differentiation of X2 cells into Xins cells.

We thank the reviewer for this insightful suggestion. We have now further tested our interpretation that *apob* orthologs are required for differentiation directly (new Fig. 6) using anti-PIWI-1 labeling of differentiating progeny in the epidermal lineage and analysis of mature epidermal cell numbers, as well as flow cytometric quantification of progeny expressing low levels of PIWI-1 protein. The results of these experiments have considerably improved the manuscript.

First, we would like to point out that our original interpretation (lines 362-364 and Fig. 6 in the first manuscript) was that ApoB inhibition caused accumulation of differentiating progeny and delays in their maturation, based on an increase in irradiation-insensitive fraction of the X2 flow cytometry subpopulation (Fig. 4). This result was further supported by the preferential dysregulation of transcripts enriched in progeny and mature cell states in the eight major planarian cell lineages (Fig. 5 and Supp. Figs. 6 & 8), and also by decreases in specific cell types (cells expressing polarity cues) and reduced organ size during regeneration (Fig. 4). Nonetheless, the reviewer's point is well-taken, that more direct evidence of delayed differentiation (especially in uninjured animals) would considerably strengthen our interpretation.

The reviewer's suggestions, to utilize EdU pulse-chase and to analyze a well-characterized lineage, have been used successfully to characterize knockdown phenotypes for genes that definitively block differentiation in specific planarian lineages. However, our data suggested an early delay in differentiation (not a block) after initial neoblast specification, with accompanying dysregulation of gene expression along each lineage trajectory. This more subtle phenotype

makes EdU pulse-chase labeling and marker-based lineage analysis problematic for a couple of reasons.

As discussed briefly above, EdU-positive progeny will be labeled along the entirety of each lineage. However, the stages at which progeny integrate into tissues (e.g., brain, pharynx, or intestine) or transition to the "Xins" gate, with respect to cell states defined by transcript expression, is not well characterized for most lineages (except perhaps the epidermis, below). Thus, we might observe either similar, or fewer EdU-positive progeny in a particular mature tissue, depending on how early in a lineage integration occurs. Therefore, interpretation of EdU pulse-chase labeling results is difficult without prior knowledge of each lineage's differentiation kinetics and testing of many lineage-specific progeny markers.

Similarly, although others have used lineage- and state-specific markers to characterize blocks in differentiation, our RNA-Seq data also raise questions about the feasibility of this approach, because we observe both up- and down-regulation of lineage-specific markers in *apob(RNAi)* animals (Fig. 5 and Supplementary Fig. 8). This obviously makes counting cells in specific states using individual transcripts alone highly dependent on the choice of marker. Consequently, such an approach might necessitate analysis of many markers, since up- or down-regulation may or may not reflect actual cell number.

For these reasons, we chose what we felt would be the **most direct and straightforward approach**: quantification of neoblast progeny expressing low levels of the PIWI-1 protein. We have added Fig. 6 describing these results, a new Results section (lines 393-450), and edited Discussion of these data (lines 456-459).

PIWI-1 protein is expressed at high levels in proliferating neoblasts, but is gradually lost in differentiating progeny, until becoming undetectable in late progeny or mature cells (Guo et al., *Dev. Cell*, 2006, DOI: 10.1016/j.devcel.2006.06.004). The exact kinetics of PIWI-1 perdurance have not been well described for any lineage, but our analysis of signature transcripts in bulk RNA-sequencing of the PIWI-1^{low} cell fraction by Zeng and colleagues (*Cell*, 2018, DOI: 10.1016/j.cell.2018.05.006) (Supplementary Figs. 7-8) supported the idea that the PIWI-1^{low} fraction was highly enriched for neoblast progeny, especially in the epidermal lineage (Supplementary Fig. 7i).

We therefore raised a PIWI-1 antibody, and focused on the well-characterized epidermal lineage, as the reviewer suggested. We hypothesized that we might observe (a) more early progeny (expressing *prog-1* mRNA, a widely used marker), and/or (b) more *prog-1*-positive early progeny expressing PIWI-1, reflecting their slower differentiation.

To test these ideas, we developed an automated image analysis method for planarian tissue sections labeled with an antisense *prog-1* riboprobe and anti-PIWI-1 (Fig. 6a-b). We did not detect significant changes in the percentage of DAPI-positive cells that also expressed *prog-1* (Fig. 6c). However, the percentage of *prog-1*+ cells that expressed low levels of PIWI ("PIWI-LO") was significantly higher in *apob-M* animals, and was modestly (but not significantly) higher

in *apob-S* animals (Fig. 6d). We also observed an upward (but not significant) trend in the much smaller fraction of *prog-1+* cells that still expressed high PIWI-1 levels ("PIWI-HI") (Fig. 6e). Additionally, the percentage of *prog-1+* cells that were PIWI-negative (e.g. late progeny and mature cells) was reduced (significantly in *apob-M* animals) (Fig. 6f). Together, these data indicated a delay in differentiation of early epidermal progeny (defined by the transition from higher to lower PIWI-1 expression). Furthermore, we also found a modest reduction in mature ventral epidermal cells (Fig. 6g) in *apob-S* animals, suggesting that the early delays correlate with less efficient production of fully differentiated progeny in uninjured animals as the *apob* RNAi phenotype becomes more severe.

Next, rather than analyzing additional lineages, we chose to use flow cytometry to quantify PIWI-LO progeny globally (Fig. 6h-k), as we thought this would be a more accurate and efficient way to assess differentiation in *apob(RNAi)* animals, without needing to identify or test state-specific markers for each lineage. As expected, and consistent with our analysis of the epidermal lineage, we found significant increases in the fraction of PIWI-LO cells in both *apob-M* and *apob-S* animals (Fig. 6j), definitively and directly showing a delay in differentiation of neoblast progeny.

In our flow cytometry experiments, we also found that the PIWI-HI fraction was modestly reduced (Fig. 6k). On further inspection, we found evidence that more neoblasts may have 2C or <4C DNA content, suggesting that putative G1/S fractions were larger relative to G2/M fractions (Supplementary Fig. 9). These data suggest that ApoB inhibition might also slow cell cycle progression, contributing to the *apob(RNAi)* phenotype. While this is interesting, we contend that delayed differentiation is likely to be the primary consequence of ApoB inhibition, because (a) we did not observe dramatic differences in the number or location of neoblasts labeled *in vivo* with stem cell markers (*piwi-1* and *tgs-1*), (b) gene expression in neoblasts was affected less than in any other planarian cell state, and (c) we did not observe enrichment of cell cycle- or pluripotency-related Biological Process GO terms among dysregulated genes in *apob(RNAi)* animals.

In summary, addressing this point has considerably strengthened the manuscript, by providing substantial, additional direct evidence that ApoB is required for differentiation. We thank Reviewer 2 again for this constructive and helpful suggestion.

Minor Comments:

1. As data demonstrates that *apolipoprotein b* orthologs are required for differentiation of "stem cell progeny to mature cell" and not for differentiation of "stem cell to progeny," several statements (mentioned below) in the manuscripts needs to be modified appropriately to avoid possible confusion.

Title: Instead of saying "...stem cell differentiation", it would be more appropriate to say "...stem cell progeny differentiation"

Line number 36: Please mention "neoblast progeny differentiation" instead of "neoblast differentiation"

Line number 100: Please mention "stem cell progeny differentiation" instead of "stem cell differentiation"

Line number 358: "ApoB was required for differentiation of planarian neoblasts" should be "ApoB was required for differentiation of planarian neoblast progeny"

Line number 361: "...planarian neoblast differentiation" should be "...planarian neoblast progeny differentiation"

Line number 363: "...differentiation of neoblast is delayed" should be "...differentiation of neoblast progeny is delayed"

We agree with the reviewer that this language is more precise and correct. We have edited the text in the locations the reviewer mentioned (lines 46, 110, 391, 454, and 457 in the revised manuscript). We have also edited the title of our manuscript to read: "Intestine-enriched *apolipoprotein b* orthologs are required for stem cell progeny differentiation and regeneration in planarians."

2. Define "*apob-M*" and "*apob-S*" at its first use. These abbreviations are first used in Fig 2i and 2j but not defined there.

We thank the reviewer for pointing out this oversight. We first introduced the "mild" and "severe" phenotypes when describing them in Fig. 2a, but we did not introduce the abbreviations until describing Fig. 3b in the Results at lines 203-204 (original manuscript).

We have added these abbreviations to the live animal phenotype descriptions in Fig. 2a, to the live regenerate phenotype descriptions in Fig. 3b, and we have edited the annotations in Figs. 2b-e for uniformity. We have also added text earlier in the results (lines 143-146, revised manuscript):

For brevity, hereafter we refer to *apob-1;apob-2(RNAi)* double knockdowns as "*apob(RNAi)*", and to specific phenotypic classes as "*apob-M*" for *apob-1;apob-2(RNAi)*-*"mild"*, and "*apob-S*" for *apob-1;apob-2(RNAi)*-*"severe"*.

This necessitated deletion of duplicate information at line 156 (revised manuscript). We did not remove a second reminder of these abbreviations in results at lines 223-224 (revised manuscript). Additionally, in response to Reviewer 3, we now add brief reminders of these abbreviations in all relevant figure legends.

3. Fig. 2f: Does Xins contains "fully differentiated/mature" cells of "late/mature progeny" as mentioned in the figure, or it's a mixture of both? Also, please label "S/G2/M" cell population as "X1".

We agree that our schematics should state the composition of the Xins fraction more precisely. Xins primarily contains fully differentiated/mature cells, but also an unknown percentage of late progeny. Multiple studies have demonstrated that some late progeny-specific markers do not decrease until 4+ days after lethal irradiation, well after significant decreases in the X1 and X2 fractions are first observed (Eisenhoffer et al., 2008, doi: 10.1016/j.stem.2008.07.002; Tu et al., 2015, doi: 10.7554/eLife.10501; and Zhu et al., 2015, doi: 10.7554/eLife.07025). Furthermore, at least one study has shown that an unknown percentage of cells in the Xins gate express these markers (Eisenhoffer et al., 2008, doi: 10.1016/j.stem.2008.07.002). We also note here that these results are consistent with our own cross-referencing of Xins "signature" transcripts with single cell profiling data (Supplementary Figures 7 and 8), in which a significant percentage of Xins-associated transcripts are also expressed by cells in progeny ("P") states in 7/8 lineages.

We have added the Eisenhoffer et al., 2008 reference at the end of the sentence (line 175) to support our statement that this fraction consists of both later stage progeny and mature differentiated cells.

We have indicated that "Xins" is comprised of both "late progeny & mature cells" in Figs. 2f and Fig. 4c, and we have also labeled the S/G2/M population as "X1," and the G0/G1 neoblast population as "X2," as the reviewer suggested, to describe these fractions more precisely in the schematics.

4. Line number 170-173: I don't see an apparent up-regulation at neoblast proliferation (1-3 days) time points in Supp Fig 2a and b. The graphs will be clearer if the authors mention asterisks at places where there is a significant fold change.

We agree with the reviewer that this section of the text did not clearly indicate which expression values were significant. In addition, our conclusion that *apob-1* and *apob-2* are upregulated at "two distinct time points" was based on statistically insignificant trends and is also not strongly supported by RNA-Seq data.

In our qPCR experiments, we observed upward trends in expression of *apob-1* and *apob-2* mRNA. However, perhaps due to low sample size and/or real biological variability at several time points (indicated in the figure), the only significant fold-change was for *apob-1* in 4 day head fragments, which was indicated with an asterisk in the original Supplementary Fig. 2a.

However, *apob-1* and *apob-2* upregulation observed in the whole-body regeneration RNA-Seq dataset that we analyzed (Zeng et al., *Cell*, 2018, doi: 10.1016/j.cell.2018.05.006) was significant (FDR-adjusted *p* value < .05) at multiple time points (Supplementary Fig. 2c). Normalization for this dataset is conducted using all transcripts, rather than a small number of

endogenous "housekeeping" control genes (Robinson et al., *Bioinformatics*, 2010, doi: 10.1093/bioinformatics/btp616). Therefore, we believe these expression values, together with the upward trends observed by qPCR provide strong evidence of upregulation during stages of planarian regeneration associated with proliferation and differentiation.

Accordingly, we have added a new line plot for the RNA-Seq data with asterisks representing significance levels to illustrate more clearly how *apob-1* and *apob-2* expression change during regeneration as part of Supplementary Fig. 2c. (We note here also that we had mistakenly inverted the original heat map; this has also been corrected.) We have also edited the Supplementary Fig. 2a and 2b legends to indicate that the upregulation was a trend but that only upregulation in head fragments at the 4 day time point was significant. Finally, we have edited the text to point out that the upregulation we observe by qPCR was only a trend, but that changes in the RNA-Seq data were significant (lines 185-190):

Using quantitative PCR, we found that both *apob-1* and *apob-2* transcripts trended upwards in tissue fragments during earlier stages of regeneration commonly associated with neoblast proliferation (1-3 days) and differentiation (2-7 days) (Supplementary Fig. 2a, b)^{48,51,52}. Significant upregulation of *apob-1* and *apob-2* was also observed in previously published RNA-Seq data from a 14-day time course of whole-body planarian regeneration (Supplementary Fig. 2c)⁵⁰.

5. Line number 175-176: Is the increase at Day 3 significant? Again, mentioning asterisk at places where there is a significant fold change will help the reader.

We observed modest upregulation at 3 days, particularly for trunk fragments, but the reviewer is correct that this upregulation was not statistically significant. Similar to our qPCR data, this was potentially due to low sample size or real variability at other time points. We have adjusted the text in the figure legend to state that we observed ApoB-1 upregulation at "4-5 dpa", and we have edited the text in the manuscript as follows (lines 190-191):

Using quantitative capillary-based Western blotting, we also found that ApoB-1 protein levels increased significantly by 4-5 days after amputation (Supplementary Fig. 2d).

6. SuppFig4B: It would be more appropriate to say "pluripotent neoblasts" rather than "totipotent neoblasts".

We agree, since the potency of the *tgs-1+* subpopulation has not been rigorously tested. Consistently, we referred to these neoblasts as "more pluripotent" in the text (line 256). We have edited the heading in Supp. Fig. 4b to "pluripotent," as the reviewer suggested.

7. Line number 399-403: The sentence needs to be rephrased as this is just speculation, and there is no direct evidence.

We agree, and we have edited this sentence to read (lines 494-498):

Because *apob* RNAi results in widespread dysregulation of thousands of transcripts associated with differentiating progeny, it is reasonable to **speculate** that in planarians, intestinal lipid stores serve as a ready carbon source that is trafficked by ApoB-containing LPs to neoblasts and progeny to support epigenetic modifications required for differentiation.

Reviewer 3 (Remarks to Authors)

Summary: An extensive literature demonstrates the importance of lipid metabolism in regulating stem cell biology. However, the role of lipid metabolism during regeneration, especially whole body regeneration, remains largely unexplored. In this study, Wong and co-authors explore the role of lipid metabolism in neoblast function during planarian regeneration. Previous work from this group demonstrated that knocking down a transcription factor (Nkx2.2) expressed in the intestine inhibited the formation of the blastema during regeneration. This intriguing result suggested that the intestine is playing an unknown role in regulating neoblasts. To identify possible mechanisms, the authors performed RNA-seq and found that two orthologs of *apolipoprotein b* (*apob-1* and *apob-2*) is highly down regulated in Nkx2.2 knockdown animals. Apolipoproteins mediate trafficking of NLs to LPs from digestive/storage organs to other tissues, thus suggesting a role for apoproteins in providing neoblasts with lipid during regeneration. Knocking down both *apob* isoforms led to a range of phenotypes, including head regression – this is indicative of a neoblast defect. The authors made a planarian specific antibody for ApoB-1 protein and found that although the *apob* mRNA is found almost entirely in the intestine, the ApoB-1 protein is found outside of the intestine, which is consistent with a conserved function for ApoB in lipid transport. Furthermore, in *apob* RNAi animals, the neoblasts had a lower NL content, suggesting that ApoB functions to transport lipids from the intestine to neoblasts. During regeneration, the *apob* genes as well as other genes involved in lipid metabolism are upregulated and *apob* RNAi led to regeneration delays and defects, which are likely explained by a delay or defect in neoblast differentiation. Accordingly, RNA-seq of *apob* RNAi animals demonstrated that differentially expressed genes were enriched for transcripts found in differentiating neoblast progeny and all lineages were affected. The authors speculate that lipids provided to neoblasts by ApoB could be a carbon source for the epigenetic modifications that are required for large scale neoblast differentiation and/or the lipids could be used an energy source.

This study is carefully done, well-written, and well presented. The authors exploration of lipid metabolism during planarian regeneration is novel and important. I have only a few very minor comments.

Authors' Response to Reviewer 3

We are very grateful to the reviewer for their positive comments and helpful advice to improve the manuscript. We have addressed the reviewer's specific suggestions below.

1. Was the antibody designed to only recognize ApoB-1 specifically, or is it expected to cross react with ApoB-2? This should be better explained, at least in the methods.

We agree that the likely specificity of the anti-ApoB-1 antibody could have been explained more clearly. We did not test specificity on protein lysates from single *apob-1(RNAi)* or *apob-2(RNAi)* knockdown animals (only the double knockdowns). However, the low similarity of ApoB-1 and ApoB-2 in this region, and the very different predicted molecular masses of the two proteins suggests that anti-ApoB-1 cross reacts minimally with ApoB-2. In addition, although we have not included the data in our manuscript, a second anti-ApoB-1 polyclonal antibody generated independently in a second rabbit recognized the same single band. We have added the following text to the Methods, lines 660-663:

The predicted N-terminus of ApoB-1 is only 22% identical and 42% similar to ApoB-2. Furthermore, ApoB-1's mass is predicted to be 235 kD less than ApoB-2. Anti-ApoB-1 labeled only a single prominent band by Western analysis (Fig. 1), suggesting the polyclonal antibody is specific for ApoB-1.

2. The meaning of ApoB-S and ApoB-M should be explained in the figure legends.

We agree that the abbreviations should be explained earlier and wherever they are used. In the original manuscript, we introduced the “mild” and “severe” phenotypes when describing them in Fig. 2a (lines 131-132), but we did not introduce the abbreviations until Fig. 3b and lines 203-204. Accordingly, we have added this text earlier in the results (lines 143-146):

For brevity, hereafter we refer to *apob-1;apob-2(RNAi)* double knockdowns as “*apob(RNAi)*”, and to specific phenotypic classes as “*apob-M*” for *apob-1;apob-2(RNAi)*-“*mild*”, and “*apob-S*” for *apob-1;apob-2(RNAi)*-“*severe*”.

We also add text clarifying these abbreviations to Fig. 2a and Fig. 3b, and in all relevant figure and supplementary data legends (Figs. 2-6; Supplementary Figs. 2, 4-6, 8, and 10; and Supplementary Data 4-5).

Reviewer Figure 1. Lipoprotein receptors are expressed in multiple tissues at multiple regeneration timepoints. (a-b) UMAP plots of ~300,000 cells from different tissues, time points, and irradiated samples, plotted together. **(a)** UMAP plot of ~300,000 planarian cells, colored by tissue (stem cells are gray, in the center). **(b)** UMAP plot of ~300,000 planarian cells, colored by regeneration time point (red to violet “ROYGBIV” = early to late = Day 0 “D00” to Day 14 “D14”). **(c-e)** Lipoprotein receptor mRNA expression (green/blue) across tissues (muscle, neurons, pharynx, etc.) and timepoints. Plots generated at https://simrcompbio.shinyapps.io/bbp_app/.

a 6dpa Tail Regenerate
Pre-pharyngeal Intestine Region

Reviewer Figure 2.

Reviewer Figure 2. Regenerating intestine of *apob(RNAi)* has similar number of cells but higher percentages of PIWI-LO and PIWI-HI cells suggesting delay in progeny differentiation.

(a-h) Quantification of DAPI+ and EdU+ cells with PIWI-1 labeling in the anterior regenerating intestine of 6 dpa tail fragments. One dpa regenerates were soaked in EdU for 6 hrs and fixed at 6 dpa. **(a)** Sagittal sections near the midline of a tail regenerate showing labeling of muscle, DAPI, EdU, and PIWI-1 in the regenerating anterior intestine. A custom FIJI macro generated the cell region masks (based on merged DAPI and EdU channels) for automated quantification of EdU and PIWI-1 signal. **(b)** Percentage of DAPI+ intestinal cells that were EdU+ in control and *apob(RNAi)* regenerates. **(c)** Number of DAPI+ cells per area (mm^2) in the regenerating intestines of control and *apob(RNAi)* animals. **(d)** Sagittal section near the midline of a 6 dpa tail regenerate outlining the pre-pharyngeal region of interest (ROI) in dashed yellow line and intestinal ROI in solid blue line. Displayed exposure is lower relative to panels in **(a)** to minimize saturation in non-intestinal regions. **(e)** Number of DAPI+ intestinal cells per area (mm^2) of the pre-pharyngeal ROI. **(f-h)** Percentages of EdU+ intestinal cells with low ("PIWI-LO") **(f)**, high ("PIWI-HI") **(g)**, or no ("PIWI-NEG") **(h)** PIWI-1 labeling. **(i-k)** Quantification of all DAPI+ intestinal cells with PIWI-1 labeling. Percentages of PIWI-LO cells **(i)**, PIWI-HI cells **(j)**, and PIWI-NEG cells **(k)** in the regenerating intestines of control and *apob(RNAi)* animals. One-way ANOVA and Tukey's multiple comparison test. Error bars: mean \pm S.D., n=3 per condition **(b-h)**. Scale bars: **(a)** 25 μm , **(d)** 100 μm .

Methods for Reviewer Figure 2. We amputated control and *apob(RNAi)* planarians seven days after the last dsRNA feeding, soaked the tail fragments in F-*ara*-EdU (for six hours) at 24 hours after amputation as in Bohr et al., *eLife*, 2021 (DOI: 10.7554/eLife.68830), and then fixed at 6 dpa, as we did in Fig. 3g-h in the manuscript. We then cryosectioned these samples and labeled EdU-positive cells as in Bohr et al., *eLife*, 2021, along with both anti-PIWI-1 and an anti-muscle antibody to distinguish the intestine boundary. Then we imaged sections (3 sections per animal from 3 animals per condition), and developed an automated quantification script in FIJI to quantify EdU and PIWI-1 levels in DAPI+ cells within the pre-pharyngeal (anterior) intestine, which regenerates by both remodeling and differentiation (Fig. 3h and Forsthoefel et al., *Dev. Biol.*, 2011). The FIJI approach was similar to that used in Fig. 6 of the manuscript (please see Methods), with two exceptions. First, an intestine-only region was first generated (lower four panels in Reviewer Fig. 2a) based on enteric muscle labeling of the outer intestinal boundary. Second, the resulting image was despeckled, smoothed, and background subtracted prior to auto local thresholding using the Phansalkar algorithm.

List of Figure and Data Edits

Figure 1. No changes.

Figure 2a. Added "*apob-M*" and "*apob-S*" abbreviations.

Figure 2b. Edited to use "*apob-M*" and "*apob-S*" abbreviations.

Figure 2c. Edited to use "*apob-VS*" abbreviation.

Figure 2d. Added dashed outlines of boundaries between intestine and parenchyma.

Figure 2e. Edited to use "*apob-M*" abbreviation.

Figure 2f. Relabeled schematic: neoblasts as X1 (S/G2) and X2 (G0/G1) populations, labeled Xins as "late progeny and mature cells".

Figure 3b. Added "*apob-M*" and "*apob-S*" abbreviations.

Figure 4c. Relabeled schematic: neoblasts as X1 (S/G2) and X2 (G0/G1) populations, labeled Xins as "late progeny and mature cells".

Figure 5. No changes.

Figure 6. New figure in revised manuscript.

Figure 7. Was Fig. 6 in original manuscript.

Supp. Fig. 1. No changes.

Supp. Fig. 2c. Re-oriented heat map. Added line plot for RNA-Seq data.

Supp. Fig. 2e-f. Added dashed outline of boundaries between intestine and parenchyma.

Supp. Fig. 3. No changes.

Supp. Fig. 4b. Changed "totipotent" to "pluripotent."

Supp. Fig. 5. No changes.

Supp. Fig. 6. No changes.

Supp. Fig. 7. No changes.

Supp. Fig. 8. No changes.

Supp. Fig. 9. New figure in revised manuscript.

Supp. Fig. 10. Was Fig. 9 in original manuscript. Added gating strategies for fixed cell flow cytometry.

Supplementary Data 6. Added information for the *prog-1* clone.

Source Data 2. Added data and statistics details for Figure 6 and Supplementary Figure 9.

REVIEWERS' COMMENTS

Reviewer #1 (Remarks to the Author):

I read the revised manuscript. As I previously said that it lacks mechanism and maybe not ready for publication on a major journal. However, I don't against its publication on Nat Commun. at current form if the editor and the other two reviewers support its publication.

Reviewer #2 (Remarks to the Author):

The revised manuscript by Wong et al. is substantially improved. This study demonstrates the role of lipid metabolism in regeneration and establishes planarians as a suitable model to understand the same. The authors have adequately addressed all of my comments, and I congratulate them on this novel and well-executed study.

To all Reviewers

We are pleased and grateful that both reviewers were satisfied with our responses to their constructive criticisms.

Reviewer #1 (Remarks to the Authors)

I read the revised manuscript. As I previously said that it lacks mechanism and maybe not ready for publication on a major journal. However, I don't against its publication on Nat Commun. at current form if the editor and the other two reviewers support its publication.

Authors' Response

We thank the reviewer for their efforts in evaluating our manuscript and for supporting its publication.

Reviewer #2 (Remarks to the Authors)

**The revised manuscript by Wong et al. is substantially improved.
This study demonstrates the role of lipid metabolism in regeneration and establishes planarians as a suitable model to understand the same.
The authors have adequately addressed all of my comments, and I congratulate them on this novel and well-executed study.**

Authors' Response

We thank this reviewer for their thoughtful evaluation, insightful suggestions, and helpful comments.